# Fast and Expressive Gesture Recognition using a Combination-Homomorphic Electromyogram Encoder

**Niklas Smedemark-Margulies**[1]                    *smedemark-margulie.n@northeastern.edu*

**Yunus Bicer**[2]                                   *bicer.y@northeastern.edu*

**Elifnur Sunger**[2]                                *sunger.e@northeastern.edu*

**Tales Imbiriba**[2]                                *t.imbiriba@northeastern.edu*

**Eugene Tunik**[2]                                  *e.tunik@northeastern.edu*

**Deniz Erdogmus**[2]                                *d.erdogmus@northeastern.edu*

**Mathew Yarossi**[2,3]                              *m.yarossi@northeastern.edu*

**Robin Walters**[1]                                 *r.walters@northeastern.edu*

[1]*Khoury College of Computer Sciences, Northeastern University*
[2]*Department of Electrical and Computer Engineering, Northeastern University*
[3]*Department of Physical Therapy, Movement, and Rehabilitation Sciences, Northeastern University*
**Reviewed on OpenReview:** *https://openreview.net/forum?id=j5T4pcLbcY*

## Abstract

We study the task of gesture recognition from electromyography (EMG), with the goal of enabling expressive human-computer interaction at high accuracy, while minimizing the time required for new subjects to provide calibration data. To fulfill these goals, we define combination gestures consisting of a direction component and a modifier component. New subjects only demonstrate the single component gestures and we seek to extrapolate from these to all possible single or combination gestures. We extrapolate to unseen combination gestures by combining the feature vectors of real single gestures to produce synthetic training data. This strategy allows us to provide a large and flexible gesture vocabulary, while not requiring new subjects to demonstrate combinatorially many example gestures. We pre-train an encoder and a combination operator using self-supervision, so that we can produce useful synthetic training data for unseen test subjects. To evaluate the proposed method, we collect and release a real-world EMG dataset, and measure the effect of augmented supervision against two baselines: a partially-supervised model trained with only single gesture data from the unseen subject, and a fully-supervised model trained with real single and real combination gesture data from the unseen subject. We find that the proposed method provides a dramatic improvement over the partially-supervised model, and achieves a useful classification accuracy that in some cases approaches the performance of the fully-supervised model.

## 1 Introduction

Subject transfer learning describes the technique of using pretrained models on unseen test subjects, sometimes with adaptation, and is a challenging and active topic of research for EMG and related biosignals (Hoshino et al., 2022; Bird et al., 2020; Yu et al., 2021). The difficulty of subject transfer arises from numerous sources.

Noise in EMG data can stem from the stochastic nature of the EMG signals, variability in EMG sensor placement and skin conductance, and inter-individual differences in anatomy and behaviors such as the selection, timing, and intensity of actions. Labels for supervised EMG tasks may also be noisy due to variation in task adherence (Samadani and Kulic, 2014). As a result of the performance gap between train and test subjects, it is typical in tasks like gesture recognition that unseen test subjects must provide supervised calibration data for training or fine-tuning models. This produces an undesirable trade-off; increasing the gesture vocabulary increases the expressiveness of the trained system but also increases subject time spent during calibration.

In this work, we propose a three-part strategy to address the trade-off between expressiveness and calibration time. First, we define a large gesture vocabulary as the product of two smaller, biomechanically independent subsets. Subjects are given a set of 4 *direction* gestures (`Up`, `Down`, `Left`, and `Right`) and 4 *modifier* gestures (`Thumb`, `Pinch`, `Fist`, and `Open`), and may either perform a *single* gesture chosen from the 4+4 individual options, or a *combination* gesture from the $4 \times 4$ options consisting of one direction and one modifier.

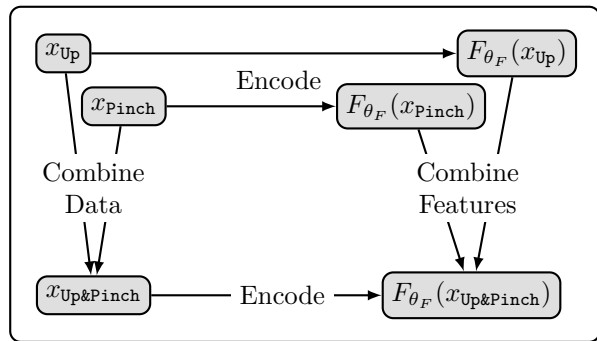

Using a gesture vocabulary with this combinatorial structure makes the system expressive, but traditional machine learning approaches cannot naively generalize to unseen combinations, and would therefore require a correspondingly large training dataset. For example, given only examples of single gestures `Up` and `Pinch`, traditional models would not naively be able to classify examples of the unseen combination `Up&Pinch` gestures. Therefore, the second key

Figure 1: Overview of desired commutativity. A combination-homomorphic encoder $F_{\theta_F}$ commutes with gesture combination: given data from real single gestures $x_{\texttt{Up}}$ and $x_{\texttt{Pinch}}$ and a real gesture combination $x_{\texttt{up\&pinch}}$, the real combination gesture's encoding $F_{\theta_F}(x_{\texttt{Up\&Pinch}})$ should equal the combined encodings of the single gestures $Combine(F_{\theta_F}(x_{\texttt{Up}}), F_{\theta_F}(x_{\texttt{Pinch}}))$.

component is to define a training scheme using partial supervision with synthetic data augmentation. A new subject only needs to demonstrate the 4+4 single gesture classes, and a classifier can then be trained using these real single gesture examples plus synthetic combination gesture examples created by combining gestures in some feature space.

In order to enact this partially supervised training, we must be able to generate useful synthetic combination gestures. Thus, the third element of our strategy is to use contrastive learning to pre-train an EMG encoder and a combination operator, such that we can combine real single gesture examples into realistic synthetic examples of combination gestures. The goal of this contrastive learning is that the encoder should be approximately homomorphic to combination of gestures, as shown in Figure 1. In other words, the property we desire is that combining two gestures in data space (e.g. when a subject perform a simultaneous `Up&Pinch` gesture) gives approximately the same features as combining two gestures in feature space (e.g. the features of an `Up` gesture combined with the features of a `Pinch` gesture). Given this property, the synthetic gesture combinations we produce can serve as useful training data.

We evaluate our method by collecting a supervised dataset containing single and combination gestures, and performing computational experiments to understand the effect of various model design choices and hyperparameters. Note that our approach to extrapolating from partial supervision can be applied to any task in which data have multiple independent labels, but where the combination classes are not easily predicted from the single classes. Classical machine-learning methods for biosignals modeling tasks often use hand-selected features; by learning our feature space entirely from data, we remove this reliance on domain expertise. In general, hand-selected features can be a useful way to enforce strong prior assumptions about a particular domain, and may be useful in the case of limited data, whereas a data-driven approach may be beneficial as the amount of available data increases or in cases when domain assumptions may be violated.

The main contributions of our work are as follows:

- We propose a two-stage classification approach that allows us to recognize a large combinatorial set of gestures from sEMG after only collecting examples of the single component gestures. Our method uses a contrastive pre-training stage, followed by a new-subject training stage with partial-supervision and synthetic data.
- We collect and release a public dataset of real-world single and combination gestures to assess our proposed method on this novel generalization task[1].
- We conduct thorough post-hoc computational experiments to demonstrate the benefit of using synthetic supervision during the calibration stage, and to explore the sensitivity of our method to many design choices and hyperparameters[2].

## 2   Related Work

**Applications of EMG.**   The broad task of gesture recognition from EMG has been well studied. Research varies widely in terms of the specific application, and thus the exact sensing modality, the set of gestures used, and the classification techniques applied. Typical applications include augmented reality and human-computer interface, as well as prosthetic control and motor rehabilitation research (Wang et al., 2017; Sarasola-Sanz et al., 2017; Pilkar et al., 2014; Bicer et al., 2023). EMG data has also been used extensively for regression tasks beyond prosthetic applications, such as hand pose estimation (Yoshikawa et al., 2007; Quivira et al., 2018; Liu et al., 2021). These applications are well-reviewed elsewhere (Geethanjali, 2016; Yasen and Jusoh, 2019; Jaramillo-Yánez et al., 2020; Li et al., 2021).

**Feature Extraction.**   One area of focus has been the effect of feature extraction on classifier performance. Most traditional approaches to EMG gesture recognition transform raw EMG signals into low-dimensional feature vectors before classification (Nazmi et al., 2016; Phinyomark et al., 2012; 2018). Feature extraction methods can be broadly characterized into those using time-domain features, frequency-domain features and time-frequency-domain features based on the method of extraction (Nazmi et al., 2016). Numerous investigations have evaluated the best set of features to maximize classifier performance, but the hardware, tasks, and findings vary between different studies, highlighting the potential issues of using hand-engineered features (Nazmi et al., 2016). Our approach differs in that we use contrastive learning to learn a feature space with the correct structure for our classification task. This approach can be re-used more easily across different experimental setups and tasks.

**Recognizing Combination Gestures.**   Previous work on gesture combinations is relatively limited. Some work has shown high accuracy in classifying movement along multiple axes with application to prosthetic control, though this was performed using fully-supervised data (Young et al., 2012). This application has been extended by using simultaneous linear regression models for controlling prostheses along multiple degrees of freedom, again using fully-supervised training data (Hahne et al., 2018). Other research has provided models for classifying single gestures from a mixed vocabulary containing both static and dynamic gestures (Stoica et al., 2012), and models to simultaneously classify a gesture and estimate force vectors (Leone et al., 2019); by contrast, we focus only on classifying static gestures. To our knowledge, only one previous article has attempted to classify combinations of discrete gestures from training data that contains only single gestures (Smedemark-Margulies et al., 2023).

**Semi-Supervised, Contrastive, and Few-Shot Learning.**   We propose a novel approach to simultaneously resolve issues with the classification of combination gestures and provide a means for subject transfer learning. We formulate the task of calibrating models for an unseen test subject as a semi-supervised learning problem, and use a contrastive loss function to ensure that items with matching labels have similar feature vectors.

Briefly, semi-supervised learning describes the setting in which labels are available for only a subset of training examples. Models are trained with a supervised loss function for labeled examples, and an unsupervised loss function on the unlabeled examples. A wide variety of unsupervised loss functions have been studied, such as

---

[1]Dataset can be downloaded at https://zenodo.org/doi/10.5281/zenodo.10291624
[2]Code to reproduce all experiments can be found at https://github.com/nik-sm/com-hom-emg

entropy minimization (encouraging the model to make confident predictions) (Grandvalet and Bengio, 2004; Lee et al., 2013), and consistency regularization (where the model must make consistent predictions despite small perturbations on the input data) (Sohn et al., 2020). These and other techniques are well-reviewed elsewhere (Weng, 2019; Ouali et al., 2020).

Contrastive learning describes techniques for learning useful representations by mapping similar items to similar feature vectors. This can be applied in tasks with full or partial labels, or even unsupervised contexts. Items may be identified as similar based on their label (Schroff et al., 2015; Khosla et al., 2020), or by applying small perturbations to input data that are known to leave the class label unchanged (Chen et al., 2020; Grill et al., 2020). This area is also well-reviewed elsewhere (Weng, 2021; Jaiswal et al., 2020).

Our approach uses a combination of ideas from the literature on semi-supervised and contrastive learning. The key step that reduces subject calibration time is to collect only partial labels, thereby defining a semi-supervised learning problem. In order to allow models to successfully extrapolate from this partial supervision, we pretrain our models using a classic contrastive learning method called the triplet loss (Schroff et al., 2015).

Our problem statement is similar to the problem statement of a standard few-shot learning setting (see Wang et al. (2020) for review), but has several additional properties. First, in few-shot learning, models are evaluated on different "tasks," where the relationship between different tasks can be quite loose (such as image classification research in which the original dataset, and even the number and identity of class labels can vary between tasks). We are interested in the variation of behavior between subjects, so each "task" in our work represents applying a pretrained model on a certain unseen test subject. This leads to a particular relationship between different "tasks"; all the same classes are present when comparing two tasks, but data is obtained from a different subject. Second, in typical few-shot learning, the labeled items available are typically randomly sampled from the full set of possible classes. By contrast, we consider a problem with two-part labels, and consider that our partial supervision consists of only single gestures (items that use only one label component) and does not contain any combination gestures.

**Subject Transfer Learning.** Note that there are a variety of other approaches to the subject transfer learning problem, such as techniques for regularized pre-training (Smedemark-Margulies et al., 2022), domain adaptation techniques based on Riemannian geometry (Rodrigues et al., 2018), and weighted model ensembling using unsupervised similarity (Guney and Ozkan, 2023). Techniques for subject transfer in EMG and related data types such as electroencephalography (EEG) are well-reviewed elsewhere (Wan et al., 2021; Wu et al., 2023).

## 3 Methods

In Section 3.1, we define the subject transfer learning problem and provide notation including model components. In Section 3.2, we describe how feature vectors from single gestures will be combined to create synthetic features for combination gestures. In Section 3.3, we describe the first stage of our proposed method in which we pretrain an encoder and a combination operator. In Section 3.4, we describe the second stage of our method, in which we collect calibration data from an unseen subject, then use the pretrained encoder and combination operator to generate synthetic combination gesture examples and train a personalized classifier for that subject.

### 3.1 Problem Statement and Notation

In a standard classification task, we seek to learn a function that predicts the correct label $y$ for a given data example $x$, based on labeled examples provided during training. Recall that the motivation for our work is to design a model that makes use of population information collected from a set of pre-training subjects, and can be quickly calibrated on new subjects. To that end, we modify this problem formulation as follows.

Consider a dataset $\mathcal{D} = \{(x, y, s)\}$ consisting of triples of data $x \in \mathbb{R}^D$, two-part gesture labels $y = (y_{dir}, y_{mod})$, and subject identifiers $s \in \{1, \ldots, S\}$. The components of the gesture label $y$ take values as follows: $y_{dir}$ is in $\{\texttt{Up}, \texttt{Down}, \texttt{Left}, \texttt{Right}, \texttt{NoDir}\}$, where $\texttt{NoDir}$ indicates that no direction was performed, and $y_{mod}$ is in $\{\texttt{Thumb}, \texttt{Pinch}, \texttt{Fist}, \texttt{Open}, \texttt{NoMod}\}$, where $\texttt{NoMod}$ indicates that no modifier was performed.

We partition the subject ID values into disjoint sets $\mathcal{S}^{Pre}$ and $\mathcal{S}^{Eval}$. Using these sets, we divide the overall dataset into three segments:

- A pre-training segment $\mathcal{D}^{Pre}$, where $s \in \mathcal{S}^{Pre}$. This is used to learn a population model.

- A calibration segment $\mathcal{D}^{Calib}$, where $s \in \mathcal{S}^{Eval}$, and containing only single gestures ($y$ has the form $(i, \texttt{NoMod})$ or $(\texttt{NoDir}, j)$). This represents the small amount of single-gesture calibration data we can obtain for a new unseen subject and is used for further model training.

- A test segment $\mathcal{D}^{Test}$, where $s \in \mathcal{S}^{Eval}$, and containing all classes $y$. This is used to measure final model performance.

We define a two-stage architecture, consisting of an encoder $F_{\theta_F}(\cdot) : \mathbb{R}^D \to \mathbb{R}^K$ producing $K$-dimensional feature vectors $z$, and a classifier $G_{\theta_G}(\cdot) : \mathbb{R}^K \to \Delta(5) \times \Delta(5)$ mapping feature vectors to pairs of 5-dimensional probability vectors. This classifier consists of two independent components, one that predicts the distribution over direction gestures, and one that predicts the distribution over modifier gestures. This "parallel" classification strategy is adopted as the simplest way of producing a two-part label prediction, though in principle one could seek to capture correlations between the two label components. Note that there will be one classifier $G_{\theta_G}^{Pre}$ used during pretraining (which provides an auxiliary loss term), and it will be replaced by a fresh classifier $G_{\theta_G}^{Test}$ trained from scratch for the unseen test subject.

For convenience, we define several subscripts to refer to data from single gestures or combination gestures. Let $\mathcal{X}_{dir}, \mathcal{Y}_{dir}, \mathcal{Z}_{dir}$ represent data, labels, and features, respectively, from a set of single gestures with only a direction component. Such a gesture has a label of the form $(i, \texttt{NoMod})$. Likewise define $\mathcal{X}_{mod}, \mathcal{Y}_{mod}, \mathcal{Z}_{mod}$ for a set of modifier gestures, which have labels of the form $(\texttt{NoDir}, j)$. More generally, let $x_{sing}$ represent data from a single gesture (either a direction-only gesture, or a modifier-only). Finally, define $\mathcal{X}_{comb}, \mathcal{Y}_{comb}, \mathcal{Z}_{comb}$ for a set of combination gestures with labels of the form $(i, j)$ where $i \neq \texttt{NoDir}, j \neq \texttt{NoMod}$, and let $x_{comb}$ represent data from one such combination gesture.

## 3.2 Homomorphisms and Combining Feature Vectors

Given two algebraic structures $G$ and $H$, and a binary operator $\circ$ defined on both sets, a homomorphism is a map $f : G \to H$ satisfying

$$f(g_1 \circ g_2) = f(g_1) \circ f(g_2), \ \forall g_1, g_2 \in G. \tag{1}$$

This property can also be described by noting that the map $f$ commutes with the operator $\circ$ (Bronshtein and Semendyayev, 2013). As mentioned previously, we design a partially-supervised calibration in which a new subject will only provide single gesture examples and we will seek to extrapolate to the unseen combination gestures. To facilitate this extrapolation, we seek to learn an encoder and a combination operator to achieve this commutativity, as shown in Figure 1. Note that there are two types of combination to consider; a physiological process that allows subjects to plan and perform a simultaneous combination of gestures, and an explicit parametric function that we will use to combine the feature vectors of two single gestures. The first type of combination (which occurs in a subject's brain and musculature) is quite complex, and for the purposes of this work is inaccessible.

Define a combination operator $(\tilde{z}_{comb}, \tilde{y}_{comb}) = C_{\theta_C}\big((z_{dir}, y_{dir}), (z_{mod}, y_{mod})\big)$ that takes as input the features and label of a real direction gesture, and the features and label of a real modifier gesture, and produces as output a synthetic feature vector and a label. The output feature vector represents an estimate of what the corresponding simultaneous gesture combination should look like. Note that all feature vectors (from single gestures, real combination gestures, or synthetic combination gestures) have the same dimension $K$. Furthermore, note that the input label $y_{dir}$ has the form $(i, \texttt{NoMod})$, and $y_{mod}$ has the form $(\texttt{NoDir}, j)$; the output label simply uses the active component from the two inputs: $\tilde{y}_{comb} = (i, j)$.

Figure 1 describes the desired relationship between the features of real and synthetic combination gestures. If we achieve the desired homomorphism property, a synthetic feature vector created by this combination operator should be a good approximation for the features from a corresponding real, simultaneous gesture.

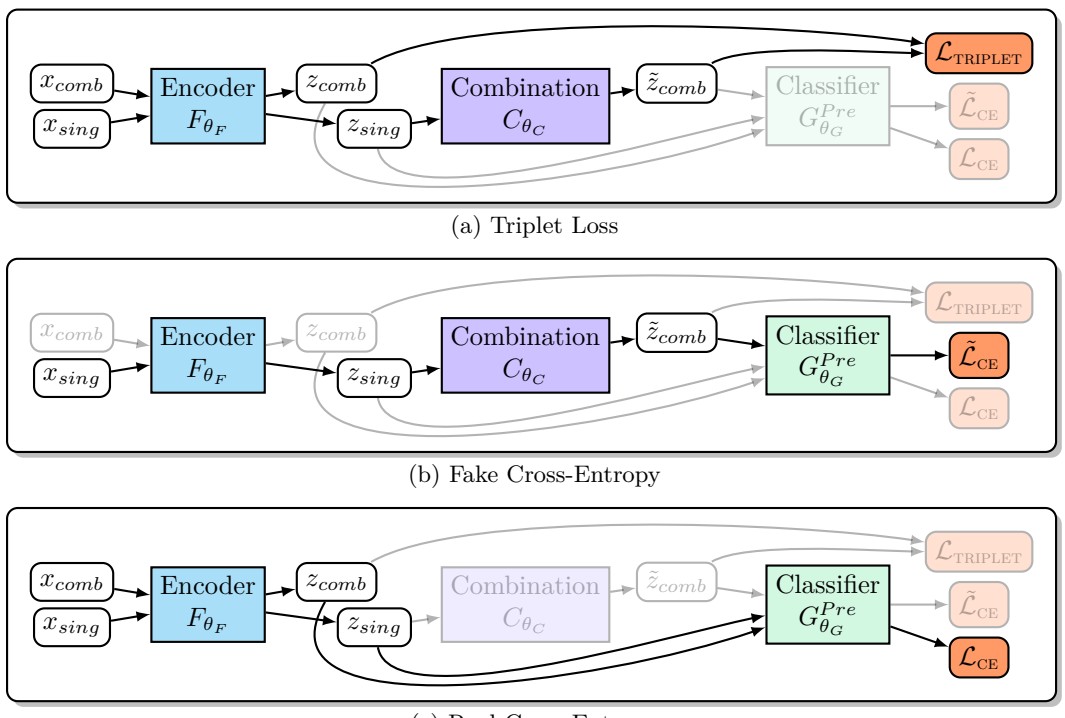

Figure 2: Fully-supervised pretraining for combination-homomorphic model. Encoder parameters $\theta_F$ and combination operator parameters $\theta_C$ are optimized using a sum of three loss terms. The classifier $G_{\theta_G}^{Pre}$ is only used for auxiliary losses, and its parameters are discarded after pretraining. Colored boxes are models; white boxes are variables; orange boxes are loss terms; inactive components are faded out. (a): The contrastive loss term $\mathcal{L}_{\text{TRIPLET}}$ compares features of real and synthetic gesture combinations. Single gestures $x_{sing}$ are encoded to features $z_{sing}$ and combined into synthetic features $\tilde{z}_{comb}$. Real combination gestures $x_{comb}$ are encoded to features $z_{comb}$. (b): Classifier $G_{\theta_G}^{Pre}$ is applied to synthetic features $\tilde{z}_{comb}$ to compute $\tilde{\mathcal{L}}_{\text{CE}}$. (c): Classifier $G_{\theta_G}^{Pre}$ is applied to real features $z_{sing}$ and $z_{comb}$ to compute $\mathcal{L}_{\text{CE}}$.

For example, given features from a single gestures $z_1$ with label $y_1 = (\texttt{Up}, \texttt{NoMod})$ and another single gesture $z_2$ with label $y_2 = (\texttt{NoDir}, \texttt{Pinch})$, the combination operator outputs a synthetic feature vector $\tilde{z}_{comb}, (\texttt{Up}, \texttt{Pinch}) = C_{\theta_C}\big((z_1, y_1), (z_2, y_2)\big)$. In the successful case, this synthetic feature vector $\tilde{z}_{comb}$ should resemble a feature vector from a real $\texttt{Up\&Pinch}$ gesture.

Note that the combination operator $C$ corresponds to the operator $\circ$ mentioned above for a general homomorphism. We consider two functional forms for $C$. In one case, we consider learning a feature space in which combination can be performed trivially. Here, $C$ has zero learnable parameters ($|\theta_C| = 0$) and computes the average of the two input feature vectors $\tilde{z}_{comb} = (z_{dir} + z_{mod})/2$, which we refer to as $C^{avg}$. This approach requires the encoder to learn a feature space in which combination occurs homogeneously for all pairs of input classes. In the other case, the combination operator $C$ produces synthetic feature output using a class-conditioned multi-layer perceptron (MLP) $\tilde{z}_{comb} = \texttt{MLP}(z_{dir}, i, z_{mod}, j)$. This allows the possibility that a flexible combination operator may still be required, and that the process of combining gestures may depend on the particular input classes.

For convenience, define another function $(\tilde{\mathcal{Z}}_{comb}, \tilde{\mathcal{Y}}_{comb}) = \texttt{CombineAllPairs}\big((\mathcal{Z}_{dir}, \mathcal{Y}_{dir}), (\mathcal{Z}_{mod}, \mathcal{Y}_{mod})\big)$ that accepts two *sets* of feature vectors with labels, and applies $C_{\theta_C}(\dots)$ to all pairs of 1 direction and 1 modifier gesture, producing a set of synthetic feature vectors with labels.

### 3.3 Pretraining the Encoder and the Combination Operator

Figure 2 describes our pretraining procedure. The goal of pretraining is to learn an encoder and feature combination operator that allows us to extrapolate from the features of two single gestures to the estimated features of their combination, as shown in Figure 1. We use contrastive learning to approximately achieve this homomorphism property. Let $d(\cdot, \cdot)$ represent a notion of distance in feature space. Consider the feature vector of a real combination gesture, such as $z_{comb} = F_{\theta_F}(x_{\texttt{up\&pinch}})$, and the feature vector of a synthetic combination gesture $\tilde{z}_{comb} = C_{\theta_C}(F_{\theta_F}(x_{\texttt{up}}), F_{\theta_F}(x_{\texttt{pinch}}))$. We seek parameters that minimize the distance between these *matching* items: $\arg\min_{\theta_F, \theta_C} d(z_{comb}, \tilde{z}_{comb})$. However, note that this does not yet provide a well-posed optimization task, since a degenerate encoder may minimize this objective by mapping all input feature vectors to a single fixed point. Thus, we must simultaneously ensure that the distance between *non-matching* pairs of items remains large; for example, the feature vector of a real combination gesture $z_{comb} = F_{\theta_F}(x_{\texttt{up\&pinch}})$ and the feature vector of a synthetic combination $\tilde{z}_{comb} = C_{\theta_C}(F_{\theta_F}(x_{\texttt{up}}), F_{\theta_F}(x_{\texttt{thumb}}))$.

We use a triplet loss function to achieve this property. Given a batch of labeled data $\mathcal{X}_{dir} \cup \mathcal{X}_{mod} \cup \mathcal{X}_{comb}$ and $\mathcal{Y}_{dir} \cup \mathcal{Y}_{mod} \cup \mathcal{Y}_{comb}$, we use the encoder $F_{\theta_F}$ to obtain features $\mathcal{Z}_{dir} \cup \mathcal{Z}_{mod} \cup \mathcal{Z}_{comb}$. We can then create feature vectors and labels for synthetic combination gestures $(\tilde{\mathcal{Z}}_{comb}, \tilde{\mathcal{Y}}_{comb}) = \texttt{CombineAllPairs}\big((\mathcal{Z}_{dir}, \mathcal{Y}_{dir}), (\mathcal{Z}_{mod}, \mathcal{Y}_{mod})\big)$. Finally, we can use a triplet loss to compare the distance between matching and non-matching combination gestures.

Consider a real combination gesture $z_{comb} \in \mathcal{Z}_{comb}$ and its label $y_{comb} = (y_i, y_j)$; this item is the "anchor." A "positive" item is a synthetic gesture $\tilde{z}_{comb}^{+}$ whose label completely matches $\tilde{y}_{comb}^{+} = (y_i, y_j)$. A "negative" item is a synthetic gesture $\tilde{z}_{comb}^{-}$ whose label differs on at least one component: $\tilde{y}_{comb}^{-} = (y_i', y_j')$ with either $y_i' \neq y_i$, or $y_j' \neq y_j$ or both. We also consider synthetic anchor items $\tilde{z}_{comb} \in \tilde{\mathcal{Z}}_{comb}$; then, positive items are *real* combination gestures with a matching label, and negative items are *real* combination gestures with a non-matching label.

Given an anchor item $a$, positive item $p$, and negative item $n$ selected as described above, and for some choice of margin parameter $\gamma$, we compute a triplet loss using

$$\mathcal{L}_{\text{TRIPLET}} = \max\left(d(a, p) - d(a, n) + \gamma, 0\right). \tag{2}$$

Intuitively, we want the negative item to be at least $\gamma$ units farther away from the anchor than the positive item.

We consider several slight variations on the standard triplet loss. In the **basic** version, for each possible anchor item in a batch (i.e. each real and each synthetic combination gesture), we select $N$ random pairs of positive and negative item, without replacement. In the **hard** version, for each possible anchor item, we form a single triplet using the farthest positive item and the closest negative item. This approach is often referred to as "hard-example mining" (Schroff et al., 2015), since these are the items whose encodings must be moved the most to achieve zero loss. In the **centroids** version, for each possible anchor item, we form a single triplet by comparing to the centroid of the positive class and the centroid of a randomly chosen negative class. At iteration $t$, given data $X_t$ from one class and a momentum parameter $M \in [0, 1]$, the centroid $C_X^{(t)}$ is computed using an exponential moving average as $C_X^{(t)} = M C_X^{(t-1)} + (1 - M)\, \mathbb{E}[X_t]$.

Note that for a fixed-sized training dataset and a sufficiently flexible encoder, there may still exist degenerate solutions in which real combinations and synthetic combinations are mapped to the same locations, but these locations are not separated in a way that permits learning a classifier with smooth decision boundaries, or does not generalize well to new subjects. Therefore, along with the encoder, we simultaneously train a small classifier network $G_{\theta_G}^{Pre}$. Using this classifier, we compute two additional loss terms; a cross-entropy on real feature vectors $\mathcal{L}_{\text{CE}}$, and a cross entropy on synthetic feature combinations $\tilde{\mathcal{L}}_{\text{CE}}$. As mentioned above, this classifier network has two independent and identical components; one is used to classify the direction part of the label, while the other classifies the modifier part. After the pretraining stage is finished, we discard the classifier; as described in the next section, we will train a fresh classifier $G_{\theta_G}^{Test}$ from scratch for the unseen subject.

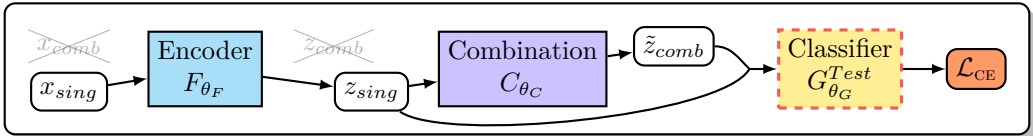

Figure 3: Partially-Supervised Calibration for Unseen Subjects. We exploit combination-homomorphism to train a fresh classifier calibration on unseen test subjects using only single gesture data $x_{sing}$ (neither real combination gestures $x_{comb}$ nor their features $z_{comb}$ are available). Data $x_{sing}$ are encoded to features $z_{sing}$ and combined into synthetic features $\tilde{z}_{comb}$. These real and synthetic gestures are pooled, and a freshly initialized classifier $G_{\theta_G}^{Test}$ is trained on these items using $\mathcal{L}_{CE}$.

Our overall training objective $\mathcal{L}$ is a mixture of the triplet loss and these two cross-entropy terms:

$$\mathcal{L} = \mathcal{L}_{\text{TRIPLET}} + \mathcal{L}_{\text{CE}} + \tilde{\mathcal{L}}_{\text{CE}}. \tag{3}$$

### 3.4 Training Classifiers with Augmented Supervision

In Figure 3, we describe our proposed method of training a fresh classifier on an unseen test subject using synthetic data augmentation. For this new test subject, we collect real labeled single-gesture data $(\mathcal{X}_{dir}, \mathcal{Y}_{dir})$ and $(\mathcal{X}_{mod}, \mathcal{Y}_{mod})$. We encode these items to obtain real feature vectors $\mathcal{Z}_{dir}$ and $\mathcal{Z}_{mod}$. We then obtain synthetic features and labels $(\tilde{\mathcal{Z}}_{comb}, \tilde{Y}_{comb}) = \texttt{CombineAllPairs}\left((\mathcal{Z}_{dir}, \mathcal{Y}_{dir}), (\mathcal{Z}_{mod}, \mathcal{Y}_{mod})\right)$. Finally, we merge the real single gesture features and synthetic combination gesture features $\mathcal{Z}_{dir} \cup \mathcal{Z}_{mod} \cup \tilde{\mathcal{Z}}_{comb}$, along with their labels $\mathcal{Y}_{dir} \cup \mathcal{Y}_{mod} \cup \tilde{\mathcal{Y}}_{comb}$ into a single dataset to train a randomly initialized classifier $G_{\theta_G}^{Test}$ for this subject. As in the pretraining stage, this classifier consists of two separate, independent copies of the same model architecture in order to classify the direction and modifier parts of the label.

## 4 Experimental Design

In this section and Section 5, we focus measuring the effect of using synthetic combination gestures during the calibration stage of our method (Figure 3). In the Appendix, we include other experiments on the effect of hyperparameters and model design choices (Section A.2) and the effect of various model ablations (Section A.3).

### 4.1 Evaluation and Baselines

For each choice of model hyperparameters, we repeat model pretraining and evaluation 50 times; this includes 10-fold cross-validation and 5 random seeds. During the calibration and test phase, the encoder is frozen, as shown in Figure 3.

As described in Section 3.1, we consider three segments of our dataset; $\mathcal{D}^{Pre}$, $\mathcal{D}^{Calib}$, and $\mathcal{D}^{Test}$. Here, we explain how the full dataset is divided into these segments. In each cross-validation fold, the pretraining dataset $\mathcal{D}^{Pre}$ contains data from 9 subjects; 1 of these 9 is used for early stopping based on validation performance. Data from a $10^{th}$ subject is divided in a stratified 80/20 split to form the calibration and test datasets $\mathcal{D}^{Calib}$ and $\mathcal{D}^{Test}$. Specifically, this 80/20 split is stratified as follows. We use the encoder to obtain real feature vectors from all of the test subject's data $\mathcal{Z}_{dir} \cup \mathcal{Z}_{mod} \cup \mathcal{Z}_{comb}$, and divide each of these portions of data to obtain a calibration dataset and a test dataset. The calibration dataset contains features and labels from direction gestures $(\mathcal{Z}_{dir}^{Calib}, \mathcal{Y}_{dir}^{Calib})$, from modifier gestures $(\mathcal{Z}_{mod}^{Calib}, \mathcal{Y}_{mod}^{Calib})$, and from combination gestures $(\mathcal{Z}_{comb}^{Calib}, \mathcal{Y}_{comb}^{Calib})$. The test dataset also contains features and labels from direction gestures $(\mathcal{Z}_{dir}^{Test}, \mathcal{Y}_{dir}^{Test})$, from modifier gestures $(\mathcal{Z}_{mod}^{Test}, \mathcal{Y}_{mod}^{Test})$, and from combination gestures $(\mathcal{Z}_{comb}^{Test}, \mathcal{Y}_{comb}^{Test})$.

Given this split of data, we can perform our augmented training scheme as described in Sec 3.4. Augmentation is performed using $\texttt{CombineAllPairs}$ with all real single features $\mathcal{Z}_{dir}^{Calib} \cup \mathcal{Z}_{mod}^{Calib}$ and their labels $\mathcal{Y}_{dir}^{Calib} \cup \mathcal{Y}_{mod}^{Calib}$. In summary, this model is trained using real single gesture items $(\mathcal{Z}_{dir}^{Calib}, \mathcal{Y}_{dir}^{Calib})$ and $(\mathcal{Z}_{mod}^{Calib}, \mathcal{Y}_{mod}^{Calib})$, as well as synthetic combination gesture items $(\tilde{\mathcal{Z}}_{comb}^{Calib}, \tilde{\mathcal{Y}}_{comb}^{Calib})$. We refer to this model as the "**augmented supervision**" model. Note that this model does not use real combination gestures $(\mathcal{Z}_{comb}^{Calib}, \mathcal{Y}_{comb}^{Calib})$.

To measure the effect of adding the synthetic gestures created using the combination operator, we compare this augmented supervision model against two baselines with different calibration data setups:

1. A "**partial supervision**" model, in which the calibration set only includes single-gesture items $(\mathcal{Z}_{dir}^{Calib}, \mathcal{Y}_{dir}^{Calib})$ and $(\mathcal{Z}_{mod}^{Calib}, \mathcal{Y}_{mod}^{Calib})$.

2. A "**full supervision**" model, in which the calibration set includes single-gesture items $(\mathcal{Z}_{dir}^{Calib}, \mathcal{Y}_{dir}^{Calib})$ and $(\mathcal{Z}_{mod}^{Calib}, \mathcal{Y}_{mod}^{Calib})$, as well as real combination items $(\mathcal{Z}_{comb}^{Calib}, \mathcal{Y}_{comb}^{Calib})$.

To ensure that our experiments can clearly show the effect of adding synthetic gesture combinations to the model calibration data, the test data is the same for all three models.

Note that these baselines are designed to measure the effect of synthetic data on model calibration. The partially-supervised baseline shows what performance we can expect if the subject demonstrates only single gestures and we naively train a classifier. It is conceivable that extrapolation from single gestures to combinations could be trivial, especially since our models contain one classifier for the direction component and one for the modifier component. In that case, the partially-supervised baseline would perform well on combination gestures, despite not having examples of them in its calibration set.

The full supervision model shows how well the classifier would have performed if we required the subject to exhaustively demonstrate all possible gesture classes (despite the burden this places on the subject). The full supervision model also demonstrates the cumulative effect of other sources of error in our experiments, such as label noise in our dataset and differences in the signal characteristics between subjects that may limit unseen subject generalization.

Note that all three models (augmented supervision, partial supervision, and full supervision) use the encoder $F_{\theta_F}$ at test time to obtain features from the unseen test subject. The baseline models do not directly use the combination operator during this calibration stage; however, the combination operator's presence during pretraining still affects the encoder's parameters, and thus indirectly affects these baselines as well.

### 4.2 Metrics: Balanced Accuracy and Feature-Space Similarity

Model performance is measured using two metrics. First, we measure the performance of the final test classifier using **balanced accuracy**, which is the arithmetic mean of accuracy on each class. Balanced accuracy is used to ensure that overall performance is not affected by any issues of class imbalance. We compute balanced accuracy on the subset of 8 single gesture classes ($Acc_{single}$), the subset of 16 combination gesture classes ($Acc_{comb}$), and the full set of 24 gesture classes ($Acc_{all}$).

Second, we characterize the **feature-space similarity** between real and synthetic gesture combinations. Recall that we pretrain the encoder $F_{\theta_F}$ and feature combination $C_{\theta_C}$ with a contrastive loss function as shown in Figure 2 in order to achieve the commutativity property shown in Figure 1. To evaluate the quality of the learned feature space, we encode all data from the unseen test subject and use a similarity metric to check whether matching classes end up being more similar to each other than non-matching classes. We use a similarity measure based on the radial basis function (RBF) kernel.

Given a pair of feature vectors $z_1, z_2$ and a lengthscale $\delta$, the RBF kernel (also called Gaussian kernel) similarity is defined as

$$K_{RBF}(z_1, z_2) = \exp(-\delta \|z_1 - z_2\|^2). \tag{4}$$

The lengthscale hyperparameter $\delta$ strongly affects the ability to detect structure using the RBF similarity metric. Using an adaptive length scale such as the median heuristic (Garreau et al., 2017) can help reveal structure within each run, but prevents the comparison of similarity values *across* runs (analogous to comparing pixel brightness values between photographs taken with different contrast levels). Therefore, we use a fixed value of $\delta = 1/128$ for all experiments. This corresponds to the expected squared L2 distance between vectors drawn from a standard Gaussian in 64 dimensions (the latent dimension of our models), which may be reasonable as a highly-simplified approximation for the typical distribution of feature vectors.

To describe the similarity between two sets of items, such as items from two classes, we compute the average RBF kernel similarity for all pairs of points. Given sets of items $\mathcal{Z}_1, \mathcal{Z}_2$, we define the set similarity metric $\texttt{SetSim}(\mathcal{Z}_1, \mathcal{Z}_2)$ as

$$\texttt{SetSim}(\mathcal{Z}_1, \mathcal{Z}_2) = \frac{1}{|\mathcal{Z}_1|} \frac{1}{|\mathcal{Z}_2|} \sum_{z_1 \sim \mathcal{Z}_1} \sum_{z_2 \sim \mathcal{Z}_2} K_{RBF}(z_1, z_2). \tag{5}$$

This definition allows us to compare two different sets of items when $\mathcal{Z}_1 \neq \mathcal{Z}_2$. To compute similarity of items in the same set when $\mathcal{Z}_1 = \mathcal{Z}_2$, we use the additional constraint that the sampled items are distinct $z_1 \neq z_2$. Note that the RBF kernel similarity takes values between 0 (for items very far apart in feature space) and 1 (for items with identical feature vectors); the set similarity $\texttt{SetSim}$ takes values in the same range.

If the encoder and combination operator provide synthetic features that are a close approximation for real features, the similarity between two matching classes, such as real $\mathcal{Z}_{\texttt{Up\&Pinch}}$ and synthetic $\tilde{\mathcal{Z}}_{\texttt{Up\&Pinch}}$, should be high. Likewise, if the triplet loss successfully separates non-matching classes, then the similarity between non-matching classes, such as real $\mathcal{Z}_{\texttt{Up\&Pinch}}$ and synthetic $\tilde{\mathcal{Z}}_{\texttt{Up\&Thumb}}$, should be low.

To describe the structure of classes in feature space, we use $\texttt{SetSim}$ to compute the entries of a symmetric $32 \times 32$ matrix $M_{Sim}$, whose $i, j$ entry is the similarity between classes $\texttt{SetSim}(\mathcal{Z}_i, \mathcal{Z}_j)$. The first 16 classes are the real combination gestures, while the next 16 classes are the synthetic combination gestures. We also summarize several key regions of $M_{Sim}$. The average of the first 16 diagonal elements represents the similarity between matching real items, i.e. the compactness of these real classes. Likewise, the average of the next 16 diagonal elements represents the compactness of the synthetic classes. The average of the $16^{th}$ sub-diagonal represents the similarity between real and synthetic items that were considered matching in the contrastive learning objective (such as real $\texttt{Up\&Pinch}$ and synthetic $\texttt{Up\&Pinch}$). Finally, the average of all other below-diagonal elements represent the similarity between non-matching items.

### 4.3 Hyperparameters and Experiment Details

**Encoder Architecture.**   The encoder model $F_{\theta_F}$ is implemented in PyTorch (Paszke et al., 2019). The model architecture is a 1D convolutional network with residual connections and has 105K trainable parameters. The encoder is pre-trained for 300 epochs of gradient descent using the AdamW optimizer (Loshchilov and Hutter, 2017) with default values of $\beta_1 = 0.9$ and $\beta_2 = 0.999$ and a fixed learning rate of 0.0003. The encoder's output feature dimension $K$ is set to 64.

**Combination Operator.**   As mentioned in Section 3.2, we consider two functional forms for the feature combination operator $C_{\theta_C}$. In one form, two input feature vectors are simply averaged, and there are no parameters in $\theta_C$. In the other, $C$ is an MLP with 17K trainable parameters.

**Classifier Architecture and Algorithms.**   We consider two options for the classifier model $G_{\theta_G}^{Pre}$ that is used during the pretraining stage. In the "small" case, $G$ has a single linear layer to classify direction and a single linear layer to classifier modifier, totaling 650 trainable parameters. In the "large" case, each of these components has multiple layers, totaling 34K trainable parameters.

When training a fresh classifier $G_{\theta_G}^{Test}$ for the unseen test subject, we use Random Forest implemented in Scikit-Learn (Pedregosa et al., 2011), since this is a very simple parametric model that also trains very fast. For comparison, we also show the performance of several other simple classification strategies for a subset of encoder hyperparameter settings: k-nearest neighbors (kNN), decision trees (DT), linear discriminant analysis (LDA), and logistic regression (LogR).

**Hyperparameters.**   When performing augmented training, we first create all possible synthetic items using $\texttt{CombineAllPairs}$, and then we select a random $N = 500$ items for each of the 16 combination gesture classes. The same subset procedure is used when computing feature-space similarities.

In all experiments, we set the triplet margin parameter $\gamma = 1.0$. For our main experiments, we include all three loss terms. For ablation experiments, described below, we consider all subsets of loss terms.

We consider three variations of a triplet loss as described in Section 3.3. When using the "basic" triplet loss strategy, we sample $N = 3$ random triplets without replacement for each item. When using the "centroids" triplet loss strategy, we use a momentum value of $M = 0.9$ to update the centroids.

**Ablation Experiments and Varying Hyperparameters.** The results of our experiments with varying model hyperparameters are shown in Appendix A.2. The results of our experiments with ablated models are shown in Appendix A.3.

### 4.4 Gesture Combinations Dataset

EMG data during gesture formation were collected from 10 subjects (6 female, 4 male, mean age $22.6 \pm 3.5$ years). All protocols were conducted in conformance with the Declaration of Helsinki and were approved by the Institutional Review Board of Northeastern University (IRB number 15-10-22). All subjects provided informed written consent before participating.

Briefly, subjects were seated comfortably in front of a computer screen with arms supported on arm rest, and the right forearm resting in an arm trough. Surface electromyography (sEMG, Trigno, Delsys Inc., 99.99% Ag electrodes, 1926 Hz sampling frequency, common mode rejection ratio: $> 80$ dB, built-in 20–450 Hz bandpass filter) was recorded from 8 electrodes attached to the right forearm with adhesive tape. The eight electrodes were positioned with equidistant spacing around the circumference of the forearm at a four-finger-width distance (using the subject's left hand) from the ulnar styloid. The first electrode was placed mid-line on the dorsal aspect of the forearm mid-line between the ulnar and radial styloid.

Custom software for data acquisition was developed using the LabGraph (Feng et al., 2021) Python package. A custom user interface instructed participants in how to perform each of 4 direction gestures and 4 modifier gestures. Subjects were then prompted with timing cues on a computer screen, instructing them to perform multiple repetitions for each of the 4+4 single gestures, and multiple repetitions of the $4 \times 4$ combination gestures (each combination gesture consists of one direction and one modifier simultaneously). From each gesture trial, non-overlapping windows of data were extracted to form supervised examples in the dataset; one window contains 500ms of raw sensor data at a sampling frequency of 1926Hz. After windowing trials, each subject provided a total of 584 single gesture examples (73 examples of each class) and 640 combination gesture examples (40 examples of each class). See Appendix A.1 for additional information on the dataset and collection procedure.

To help prevent overfitting, we add noise to the data during training. In each training batch, we consider each class of data present, and add freshly sampled white Gaussian noise such that the signal-to-noise (SNR) ratio of the data is roughly 20 decibels. Given the raw data items from a single class $\hat{X}$ with dimension $D$, we produce noisy data $X$ with SNR of $B$ decibels as follows:

$$X = \hat{X} + \sigma_B \varepsilon, \quad \sigma_B = \frac{\sigma_X}{(10^{B/20})}, \quad \sigma_X = \sqrt{\mathbb{E}[(\hat{X} - \mathbb{E}[\hat{X}])^2]}, \quad \varepsilon \sim \mathcal{N}(0, I_D) \tag{6}$$

No other filtering or pre-processing steps are performed. Pre-processing may provide additional benefit to the absolute performance of a gesture recognition model, though this is orthogonal to the focus of the proposed work.

## 5 Results

In our experiments, we focus on examining the effect of adding augmented supervision during the calibration stage (See Figure 3). As described previously, we compare a model that receives augmented supervision, with a baseline model that receives only partial supervision (i.e. single gesture examples only) and another baseline that receives full supervision (i.e. real single and real combination gestures).

### 5.1 Balanced Accuracy and Confusion Matrices

**Balanced Accuracy.** In Table 1, we compare the balanced accuracy of the proposed augmented training scheme to the partially-supervised and fully-supervised baseline methods. This table shows results for a single

Table 1: Balanced accuracy, for the 8 single gesture classes ($Acc_{single}$), the 16 combination gesture classes ($Acc_{comb}$), and all 24 classes ($Acc_{all}$). Entries show mean and standard deviation across 50 trials (10 data splits and 5 random seeds). The proposed augmented training scheme greatly increases overall model accuracy by trading a modest reduction in performance on single gestures for a large increase in performance on combination gestures. See Section 4.1 for baseline model details.

| Model | $Acc_{single}$ | $Acc_{comb}$ | $Acc_{all}$ |
|---|---|---|---|
| Partial Supervision | $0.90 \pm 0.05$ | $0.01 \pm 0.01$ | $0.31 \pm 0.02$ |
| Full Supervision | $0.80 \pm 0.07$ | $0.58 \pm 0.09$ | $0.65 \pm 0.07$ |
| **Augmented Supervision** | $0.77 \pm 0.11$ | $0.32 \pm 0.08$ | $0.47 \pm 0.06$ |

choice of hyperparameters that gave the highest overall accuracy and a good balance between performance on single gestures and combination gestures. For data shown here, the auxiliary classifier during pretraining was the "small" single-layer architecture; the combination operator was an MLP, and the "basic" triplet loss was used; results from all hyperparameters are shown in the Appendix in Table 3. The encoder $F_{\theta_F}$ and combination operator $C_{\theta_C}$ were pretrained as described in Section 3.3, and then a fresh classifier $G_{\theta_G}^{Test}$ was trained using different forms of supervision on the unseen test subject. In the augmented supervision case (bold row), the classifier was trained as in Section 3.4; the partially-supervised and fully-supervised baselines were trained as in Section 4.1.

Comparing the performance of the partially-supervised baseline to the augmented supervision model, we observe that adding synthetic combination gestures leads to a large improvement of about $+16\%$ overall classification performance on the unseen test subject. The overall gain is achieved by trading a small decrease in performance on the single gesture classes ($-13\%$) for a relatively larger increase in performance on the combination gesture classes ($+31\%$). The resulting model achieves performance much better than random chance on all classes, whereas the partial supervision model achieves essentially zero percent accuracy on its unseen combination gesture classes. Recall that all three models (augmented supervision, partial supervision, and full supervision) use the same parallel approach to classification; for a given gesture, one model classifies the direction component, and another independent model classifies the modifier component. If extrapolating to gesture combinations was trivial, then the partial supervision model would have enough information to predict combination gestures, since it has already seen all the individual gesture components. The fact that this partial supervision is *not* enough demonstrates that gestures compose in a highly non-trivial manner. The fact that gestures compose in a non-trivial manner is not surprising from a biological perspective (Scott and Kalaska, 1997; Liu et al., 2014); for example, the set of muscle activations required to form a `Pinch` gesture are different in a neutral wrist angle, than while simultaneously performing an `Up` or `Down` gesture.

**Trade-off of Single and Combination Performance**   The trade-off in performance between single and combination gestures is noteworthy, and is a fundamental element of the problem setting and our approach. As mentioned in Section 3.1, all models consist of two independent classifier copies, with one responsible for predicting direction and one for predicting modifier. Consider the direction classifier as an example.

In the partially-supervised case, the direction classifier is trained to predict one of 5 labels, and its training data came from 5 relatively compact classes: data that should be classified as `Up` came from gestures whose full label was (`Up,NoMod`); data that should be classified as `Down` came from (`Down, NoMod`) gestures, etc. As a result, this classifier learns specialized decision boundaries that perform well for this particular subset of the data distribution (i.e. single gesture data).

In the fully-supervised or synthetic supervision case, the direction classifier is still trained to predict one of 5 labels, but its training data come from 5 heterogeneous classes: training data for the `Up` label came from the classes (`Up,Pinch`), (`Up,Thumb`), ..., (`Up,NoMod`), and likewise for other direction labels. In this case, the classifier is forced to learn decision boundaries for a broader and more heterogeneous data distribution. The decision boundaries that capture this greater variability are worse when focusing on only the single gesture portion of the data distribution, but better when considering the full data distribution.

To explore this phenomenon, we use the performance of a small classifier as an empirical measurement of separation between different groups of items, since groups of data that are well-separated can be classified with high accuracy. For each subject, we consider three scenarios, and train a Random Forest classifier using an 80 : 20 train:test split. First, we compare single and combination data by splitting each subject's data into two classes: those with labels (`D, NoMod`) and `NoDir, M`, and those with labels (`D, M`). Next, we compare the variation of direction data by splitting each subject's data into four classes: (`Up, M`), (`Down, M`), (`Left, M`), and (`Right, M`). Lastly, we compare the variation of modifier data by splitting each subject's data into four classes: (`D, Pinch`), (`D, Thumb`), (`D, Fist`), and (`D, Open`). Table 2 shows the results of this experiment, where balanced accuracy is computed within each subject, and then averaged across subjects. We see that it is roughly as easy to distinguish single data from combination data (73.2%), as it is to distinguish between the four active direction gestures (74.7%), or to distinguish between the four active modifier gestures (66.7%). This leads to the phenomenon described above.

Table 2: Within-subject performance of Random Forest classifier on raw sEMG data for various classification tasks. Mean and standard deviation of balanced test accuracies averaged across 10 subjects and 5 random data splits. The similar classification accuracy indicates that separation of single vs combination data is roughly similar to the separation of direction or modifier classes themselves. Due to this separation, the Partial Supervision model, which is trained with only single-gesture data, becomes overly-specialized for that subset of the data distribution, and performs better on that subset of items but worse overall (See Table 1).

| Task | Balanced Test Acc (mean $\pm$ std) |
|---|---|
| Single vs Comb. | $0.732 \pm 0.060$ |
| Which Direction | $0.747 \pm 0.069$ |
| Which Modifier | $0.667 \pm 0.090$ |

**Confusion Matrices.**  Figure 4 presents the performance of the same three models in the form of confusion matrices. Here, the striking gain of performance on the combination gesture classes can be seen more clearly, as well several noteworthy patterns of errors. The dotted lines show the boundary between the single gesture classes, the combination gesture classes, and a (`NoDir, NoMod`) class. Items in the bottom-left region of a plot are combination gestures that were incorrectly classified as single gestures.

Figure 4(a) shows the performance of the partial supervision model. This model performs well on the 8 single gesture classes on the diagonal; however, almost none of the combination gesture classes are correctly classified. Instead, one of the two components is often classified as `NoDir/NoMod` resulting in an entry in the lower-left region of the matrix, or both components are classified as `NoDir/NoMod`, resulting in a prediction in the right-most column. As an example, this shows that seeing single gesture training data for (`Down, NoMod`) does not prepare the model to understand the direction component of a (`Down, Pinch`) gesture. This partially-supervised model often correctly identifies the direction component, but predicts that the modifier component is unlike its training data and belongs in the `NoMod` class.

Figure 4(b) shows the performance of the full supervision model, which achieves strong performance on all classes. Even in this fully supervised case, the strongest pattern of errors appears when the model sees a real combination gesture, and successfully identifies the direction component, but fails to identify the modifier component.

Finally, Figure 4(c) shows the performance of our proposed model trained with augmented supervision. There is a striking gain of performance on the combination gesture classes, with individual classes gaining between 9% and 55% balanced accuracy. This improvement is visible as the appearance of a strong diagonal line in the lower right region on the confusion matrix.

## 5.2 Latent Feature-Space Similarity

The core idea behind our approach is that we can extrapolate to unseen gesture combinations by using a contrastive objective to ensure that synthetic combination gestures are placed appropriately in feature space

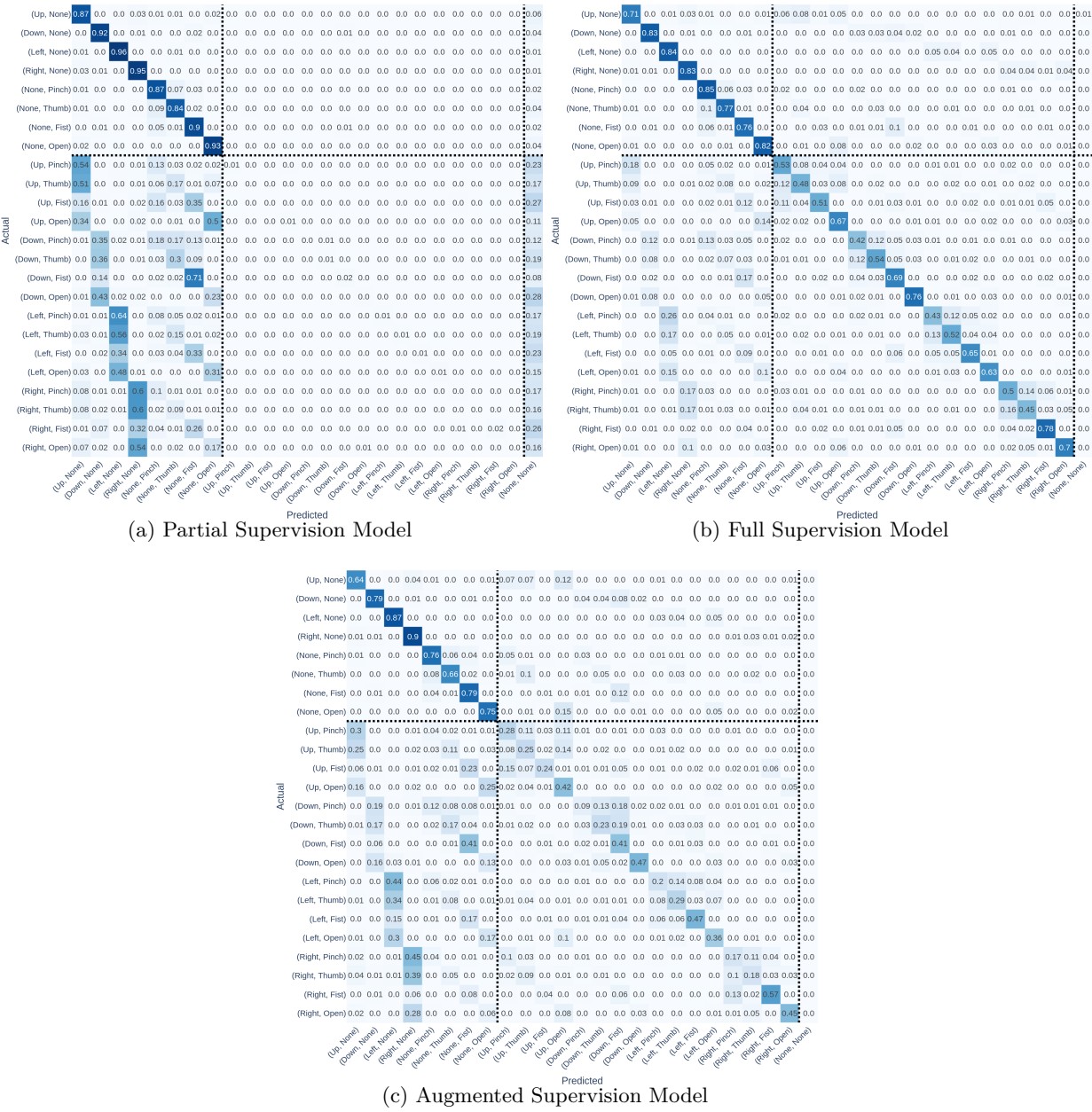

Figure 4: Confusion matrices corresponding to Table 1. The entry at row $i$ column $j$ shows the fraction of items known to belong to class $i$ that were predicted to belong to class $j$. (a) Model calibration performed using partial supervision (only single gesture examples). (b) Model calibration performed using full supervision (real single and combination gesture examples). (c) Model calibration performed using proposed method of augmented supervision. Dotted lines show the boundary between the 8 single gesture classes, the 16 combination gesture classes, and the 1 class for outliers. Whereas the partial supervision model achieves nearly zero performance on combination gestures, the augmented supervision model shows strong performance improvement for combination gestures.

during training. In Section 5.1, we examined how using these synthetic combination gestures as training data during the calibration stage affects the performance of a classifier. Here, we directly examine whether the

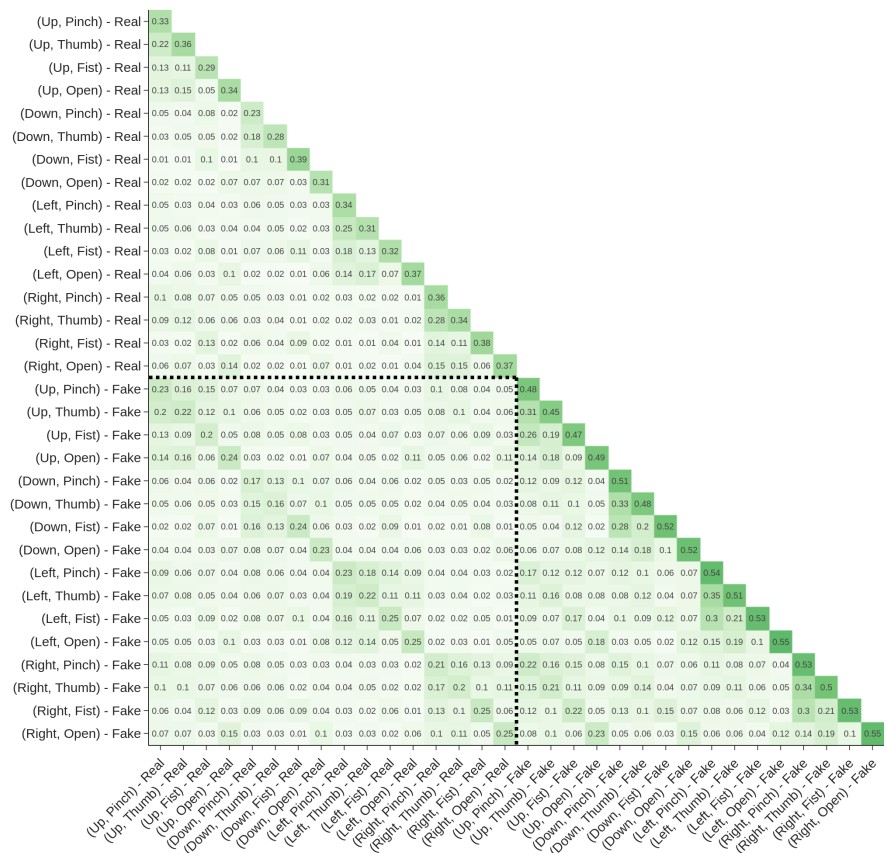

Figure 5: Feature-space similarity between real and synthetic ("fake") combination gestures. We show the lower-triangular portion of the symmetric matrix $M_{Sim}$ as described in Section 4.2. The entry at row $i$ column $j$ shows average RBF kernel similarity between items of class $i$ and class $j$ (with values ranging in $(0, 1]$. Dotted lines separate the real and synthetic classes. Similarity matrix was averaged across 50 independent runs. The main diagonal and $16^{th}$ subdiagonal are clearly visible, indicating that items with matching label are more similar than items with non-matching label.

contrastive learning procedure results in synthetic items being similar to matching real items and dissimilar from non-matching items.

Figure 5 shows the similarity matrix $M_{Sim}$ as defined in Section 4.2. Using the same model whose results are shown in Table 1 and Figure 4, we extract the features of all items from the unseen test subject, and construct 500 synthetic combination gestures per class. We compare each pair of classes using the set similarity metric SetSim, and fill in the entries of the similarity matrix $M_{Sim}$ as described in Section 4.2. Finally, this matrix is averaged across 50 independent re-runs of the training procedure (comprising 10 data splits and 5 random seeds).

In the upper-left triangle, we see the similarity between pairs of real gesture classes. The strongest entries are on the diagonal, indicating that items are more similar to other items of the same class than to items from non-matching classes. The entries in the $1^{st}$, $2^{nd}$, and $3^{rd}$ sub-diagonal show that items with the same direction component but different modifier component are also sometimes similar.

In the lower-right triangle, we see the similarity between pairs of synthetic gesture classes. The diagonal elements are again the strongest, showing that classes are relatively compact and well-separated. The range of values in the lower-right region (both diagonal and off-diagonal) is higher than values in the upper-left region; this may indicate that the synthetic combination items are more compact overall (increasing all

similarity values). Since the combination operator here was an MLP, it is not necessary for synthetic items to be closer to each other than real items; however, it is possible that this structure arose during pretraining. In particular, it is possible that the combination operator learns to place synthetic gestures close to the mean of the corresponding class, since this would lead to a better triplet loss on average.

In the bottom-left square region, we see the comparison between real and synthetic items. The diagonal is clearly visible, indicating that items with exactly matching label are most similar to each other, as desired. There is also a noteworthy block-diagonal pattern, indicating of similarity between items whose direction label matches, but whose modifier label does not match. For items whose label is totally non-matching, we see very low similarity values.

# 6 Discussion

## 6.1 Summary and Key Takeaways

We study the problem of gesture recognition from EMG, with the goal of maximizing expressiveness while minimizing the calibration time required for an unseen test subject. Typical approaches to modeling biosignals require new subjects to perform burdensome calibration sessions and exhaustively demonstrate all regions of input space, due to the large differences in signal characteristics between individuals. This exhaustive calibration scales with the number of classes and results in a large up-front time cost.

We construct a large gesture vocabulary as the product of two smaller sets; subjects may perform a single gesture or a combination gesture. We propose a two-stage method for performing classification while keeping calibration short. During fully-supervised contrastive pre-training, we train an encoder and combination function to produce useful synthetic gesture combinations from real single gesture examples. Then for an unseen test subject, we collect only single gesture examples and extrapolate to their synthetic combinations. This proposed method simultaneously keeps calibration brief, since subjects only demonstrate a linearly-sized set of single classes, while also keeping model expressiveness high, since subjects can use a quadratically-sized set of combination classes.

To evaluate the proposed method, we collect a real-world dataset of gesture combinations. Subjects follow visual prompts to perform repetitions of 4 direction and 4 modifier gestures, as well as the full set of 16 combinations; disjoint 500ms windows of 8-channel sEMG are recorded during gesture production. Our main experiments are designed to measure the effect of the proposed synthetic supervision at calibration time. We therefore compare the proposed method against two baselines: a partial supervision case in which the unseen subject's model is trained using real single gestures with no augmentation, and a full supervision case in which the unseen subject's model is provided real supervision for both single and combination gesture classes. The partial supervision model demonstrates the performance decrease we would naively suffer by collecting only single gesture examples from the unseen subject. The full supervision model demonstrates the maximal performance that we could achieve on a particular unseen test subject using a particular encoder if we did not attempt to reduce time spent collecting calibration data, considering factors such as classifier algorithm and various sources of noise in the dataset. We measure model performance using balanced accuracy on single gestures, combination gestures, and all gestures together. To evaluate the structure of the learned feature space, we also use a set-similarity metric based on an RBF kernel to measure the similarity between matching and non-matching items. We repeat all experiments across multiple random seeds and data splits.

In our main experiments, presented in Section 5.1 and 5.2, we find that the proposed synthetic supervision greatly improves classification of unseen gesture combinations ($+31\%$) at a cost of performance on single gesture classes ($-13\%$), with an overall improvement across all classes ($+16\%$). We explore this trade-off between single and combination performance further. We also find a clear pattern of feature-space similarity indicating that the pre-trained encoder and combination function are able to produce synthetic data that are well-clustered by class, well-separated, and highly similar to the corresponding real classes.

In supplementary experiments, presented in Appendices A.2 and A.3, we measure the effect of: different versions of a contrastive triplet loss, auxiliary loss terms used in pre-training, the architecture of an auxiliary

classifier used during pre-training, the classifier algorithm used during calibration, the form of the learned composition function, and the amount of noise added for regularization during training.

## 6.2 Future Directions

There are several sources of error that limit the generalization performance of our method in the current study. As mentioned, an unseen test subject may differ from training subjects due to measurement noise, differences in signal characteristics, or differences in motor behavior. The dataset also contains label noise, since labels are applied based on a visual task instruction, but subjects may perform the wrong gesture or with inaccurate timing. Furthermore, the proposed contrastive learning approach forces the model to adjust the structure of its feature space; it is conceivable that this could impede the ability to train a classifier on the unseen subject (or that the effect of the auxiliary cross-entropy loss terms during pretraining was not sufficiently strong). There are other types of modeling changes that could help increase absolute performance for the full supervision, such as feature engineering and neural architecture search; these techniques are orthogonal to the proposed strategy for augmented supervision. Future experiments using ground-truth labels of hand position may further improve the absolute performance of our method by reducing label noise.

We briefly discuss the model accuracies achieved in our experiments from the perspective of a real-world user-facing application. Note that the final balanced accuracy of our proposed model is 77% for single gestures and 32% for combination gestures (47% over all classes). For a user-facing application where very high accuracy is required, a number of standard techniques can be applied on top of such a model to achieve higher final accuracies. For example, Xu et al. (2022) develop a model for gesture recognition using inertial measurement unit signals measured at the wrist. They develop a model whose accuracy on 4 active gesture classes ranges from 62% to 74% when classifying individual data windows ("window-level," Fig 3 (a) of Xu et al. (2022)). The same model achieves an accuracy between 93% and 97% when aggregating predictions over 3 to 4 consecutive windows ("gesture-level," Fig 3 (b) of Xu et al. (2022)). Other techniques include aggregating predictions across an ensemble of independent model copies (e.g. see Du et al. (2022) for discussion of the mixture-of-experts technique), or incorporating contextual information in the form of a Bayesian prior, though this requires application-specific modeling (e.g. see Speier et al. (2016) and Orhan et al. (2012) for applications to assistive typing via electroencephalography).

One of the key design considerations of our method is the structure of the combination operator. In our main experiments, we considered a class-conditioned MLP. Providing additional information to the combination operator may be an avenue for future innovation, such as summary statistics that describe the feature-space structure of the test subject's gestures.

For the purpose of clearly measuring the effect of the proposed method, we used a simple downstream classification strategy. The relative changes in accuracy that we observe in our experiments clearly demonstrate the benefit of the proposed method. Further performance gains could be achieved by using more sophisticated downstream classification methods and other transfer learning techniques such as model ensembling or regularized pre-training. The time required for demonstrating examples could also be further reduced by using active or continual learning techniques to reduce the number of examples provided for each gesture class. We study an extreme point on the spectrum of supervision, with zero real combination gesture examples during calibration; our method could be supplemented with a few real combination examples (e.g. a linearly-sized subset of combinations) to obtain a different trade-off between performance and calibration time.

While generalizing to unseen combinations is a topic of broad interest in machine learning (e.g. Nikolaus et al. (2019); Madan et al. (2020); Wiedemer et al. (2024)), we emphasize an important aspect of our application that differs from the standard setting. In typical multi-label settings where we may study combinatorial generalization, data examples often have a structured multi-part label. Depending on the representation of input data, it is possible for one label to be unobserved and thus absent from our dataset, but that label still exists in the underlying physical model. For example, consider recognizing objects with multiple attributes such as shape and color from a text description. For some inputs, we may be missing information about shape, but that object's shape still exists. In our setting, a single gesture has a partial label, not because of missing information, but because the other label component is truly inactive.

Future experiments may also consider an expanded dataset from a similar setting. One of the simplest ways to expand the current work would be to increase the number of items in each of our single gesture sets. In our experiments, the gain for combination gesture accuracy $(+31\%)$ was much larger than the loss for single gestures $(-13\%)$, and thus we observed an overall benefit $(+16\%)$. If this approach gave similar increases and decreases on a dataset with more single gestures, the net improvement in performance across all classes would grow, since a larger fraction of the total labels would be combinations (for $N$ direction and $M$ modifier gestures, combination gestures constitute $\frac{NM}{N+M+NM}$ fraction of all classes).

Our proposed method may be modified for other applications. One area is tasks with multiple rare labels, such as classifying biomedical signals where rare disease states can co-occur. For example, consider electrocardiogram (ECG) measurement and multiple abnormal cardiac states that may be visible in ECG (e.g. see Houssein et al. (2017) for review). We may want to classify ECG signals as containing either one or multiple disease signatures, but there may be insufficient real combination training data due to the rarity of each single disease state. This application may require adjusting our approach, since we used a fully-supervised pre-training stage with real combination data.

Another area of possible application is tasks with many-part labels where exploring all combinations is not feasible, such as robotic learning and world modeling (e.g. see Muratore et al. (2022) for review). In such applications, the space of possible combinations is too large to feasibly collect real training data from all combinations. Thus, our proposed method of using synthetic supervision from unseen combinations may be useful to supplement real training data.

### Broader Impact Statement

Gesture recognition from EMG may increase the ability to interact expressively with computer systems, but the failure modes of these models may not be well-defined. Thus, gesture recognition systems should be used with care in sensitive production environments.

Some research has been done in the area of biometrics and de-anonymizing EMG and related biosignals. Extracting a subject's identity from anonymized recordings of biosignals is only possible if a reference dataset is available from that individual. If a reference signal for a certain person is available, and if they use a gesture recognition system with an expectation of anonymity, then the possibility that they may be identified from their anonymized data may lead to a violation of privacy.

### Acknowledgments

Funding for this research was provided by Meta Reality Labs Research.

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

## A  Appendix

In Section A.1, we give additional details on the paradigm used when collecting our dataset. In Section A.2, we conduct hyperparameter search experiments in which we vary model architecture, try a learned and a fixed feature combination operator, and vary the contrastive loss function. These experiments show that the proposed method is robust to these choices. The primary effect of varying these hyperparameters is to change the trade-off between performance on single and combination gestures; however the proposed augmented supervision method yields a large overall benefit in all cases. In Section A.3, we conduct ablation experiments to check the importance of each term in our pre-training objective, as well as the effect of adding noise to the raw input data during pre-training. We find that all three terms in our loss function are necessary to achieve maximal performance, and that the chosen level of input noise yields the best performance.

### A.1  Dataset Details

Our supervised dataset of single and combination gestures was obtained by recording surface EMG while subjects performed gestures according to visual cues. EMG was recorded from 8 electrodes (Trigno, Delsys Inc., 99.99% Ag electrodes, 1926 Hz sampling frequency, common mode rejection ratio: > 80 dB, built-in 20 − 450 Hz bandpass filter), spaced uniformly around the mid forearm as shown in Figure 6.

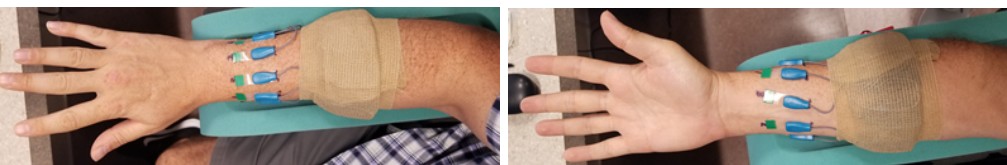

Figure 6: Surface EMG Recording setup. Electrodes were placed on the mid-forearm of the subject starting from mid-line on the dorsal aspect and continuing towards the thumb.

Subjects followed visual prompts to perform each gesture, as shown in Figure 7. The experimental paradigm consisted of 6 blocks.

In the first block, subjects were shown the UI in Figure 7a and performed single gesture trials. Each single gesture trial consisted of 3s of preparation time (indicated by a yellow screen border), followed by 2s of active time holding the gesture (green screen border), and finally 3s of rest time (red screen border). From each single gesture trial, we extracted a single 500ms window of data centered within the active period.

In the next five blocks, subjects were shown the UI in Figure 7b and performed combination gesture trials. Each combination gesture trial consisted of 2s of preparation time (yellow screen border), followed by 8s of active time (green screen border), and 2s of rest time (red screen border). The structure of each combination gesture trial was defined by two horizontal line segments. One segment (the "held" gesture) spanned the full 8s of active time, while the other (the "pulsed" gesture) was present for 4 intervals (alternating between a 1s interval, and a 0.5s interval). The vertical gray cursor gradually moved from left-to-right during the trial; subjects were instructed to follow the cursor and perform gestures as the cursor intersected with the horizontal line segments. In some trials, one of the horizontal lines was omitted; in this case, a single gesture was held for 8s, or a single gesture was pulsed 4 times.

From each combination gesture trial, we extracted 500ms windows of data as follows. Consider the arrangement of the line segments as shown in Figure 7b; there are 9 intervals of interest during the active period (5

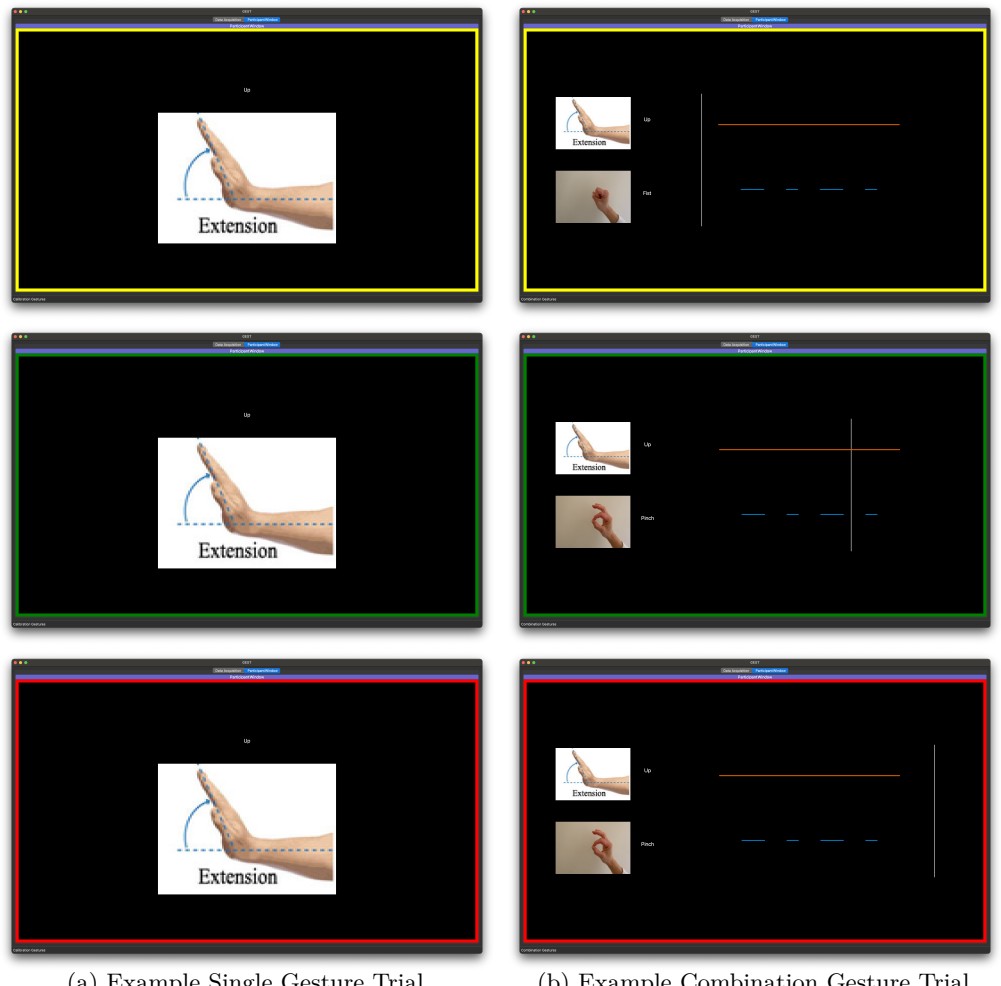

(a) Example Single Gesture Trial      (b) Example Combination Gesture Trial

Figure 7: User interface (UI) shown to subjects during supervised data collection. (a): UI during a single gesture trial. The screen border was yellow for 3s of preparation, then green for 2s of active time, then red for 3s of rest time. (b): UI during a combination gesture trial. The gray vertical cursor scrolled from left to right; when it intersected a horizontal line segment, the subject performed that gesture. The screen border color also indicated when the subject should be active (yellow for 2s of preparation, then green for 8s of activity during the trial, then red for 2 of rest).

intervals when only the held gesture is active, and 4 intervals when both gestures are active). We extracted a window of data centered in each interval, with labels decided as follows:

- In trials that contained a held gesture and a pulsed gesture (the majority of trials), we extracted 4 combination gesture windows.

- In trials that contained only a held gesture, we extracted 9 single gesture windows.

- In trials that contained only a pulsed gesture, we extracted 4 single gesture windows.

Single gesture data from all blocks was pooled, giving a total of 584 single gesture example per subject (73 examples for each of the 8 single gesture classes). Likewise, combination gesture from all blocks was pooled, giving a total of 640 combination gesture examples per subject (40 examples for each of the 16 combination gesture classes). When constructing data splits during model training and evaluation, we used a stratified split as described in Section 4.1.

## A.2 Varying Hyperparameters

### A.2.1 Effect on balanced accuracy

Table 3: Effect of varying model hyperparameters on balanced accuracy, for the 8 single gesture classes ($Acc_{single}$), the 16 combination gesture classes ($Acc_{comb}$), and all 24 classes ($Acc_{all}$). Augmented supervision; classifier for unseen subject trained according to Section 3.4. Partial Supervision and Full Supervision; baseline models described in Section 4.1. $G_{\theta_G}^{Pre}$ type; structure of auxiliary classifier used during pretraining. $C_{\theta_C}$ type; structure of combination operator. Triplet type; loss function for pretraining. See Section 4.3 for details on hyperparameter values. All models used random forest algorithm for unseen test subject. Entries show the mean and standard deviation across 50 independent trials. Bold row represents model shown in Table 1 and Figure 4.

| Hyperparameters | | | Augmented Supervision | | | Partial Supervision | | | Full Supervision | | |
|---|---|---|---|---|---|---|---|---|---|---|---|
| $G_{\theta_G}^{Pre}$ type | $C_{\theta_C}$ type | Triplet type | $Acc_{single}$ | $Acc_{comb}$ | $Acc_{all}$ | $Acc_{single}$ | $Acc_{comb}$ | $Acc_{all}$ | $Acc_{single}$ | $Acc_{comb}$ | $Acc_{all}$ |
| large | avg | basic | $0.69 \pm 0.15$ | $0.30 \pm 0.06$ | $0.43 \pm 0.06$ | $0.89 \pm 0.06$ | $0.01 \pm 0.01$ | $0.30 \pm 0.02$ | $0.79 \pm 0.08$ | $0.57 \pm 0.10$ | $0.65 \pm 0.09$ |
| large | avg | centroids | $0.68 \pm 0.13$ | $0.30 \pm 0.05$ | $0.43 \pm 0.05$ | $0.89 \pm 0.06$ | $0.01 \pm 0.01$ | $0.30 \pm 0.02$ | $0.77 \pm 0.08$ | $0.56 \pm 0.08$ | $0.63 \pm 0.08$ |
| large | avg | hard | $0.68 \pm 0.14$ | $0.31 \pm 0.07$ | $0.44 \pm 0.06$ | $0.89 \pm 0.06$ | $0.00 \pm 0.01$ | $0.30 \pm 0.02$ | $0.77 \pm 0.07$ | $0.56 \pm 0.10$ | $0.63 \pm 0.08$ |
| large | mlp | basic | $0.85 \pm 0.08$ | $0.22 \pm 0.06$ | $0.43 \pm 0.05$ | $0.90 \pm 0.05$ | $0.01 \pm 0.01$ | $0.31 \pm 0.02$ | $0.79 \pm 0.07$ | $0.58 \pm 0.10$ | $0.65 \pm 0.08$ |
| large | mlp | centroids | $0.71 \pm 0.13$ | $0.28 \pm 0.06$ | $0.42 \pm 0.04$ | $0.90 \pm 0.05$ | $0.01 \pm 0.01$ | $0.31 \pm 0.02$ | $0.79 \pm 0.07$ | $0.56 \pm 0.08$ | $0.64 \pm 0.07$ |
| large | mlp | hard | $0.84 \pm 0.06$ | $0.21 \pm 0.06$ | $0.42 \pm 0.04$ | $0.88 \pm 0.04$ | $0.01 \pm 0.01$ | $0.30 \pm 0.02$ | $0.77 \pm 0.06$ | $0.54 \pm 0.08$ | $0.62 \pm 0.07$ |
| small | avg | basic | $0.67 \pm 0.14$ | $0.38 \pm 0.07$ | $0.47 \pm 0.05$ | $0.90 \pm 0.04$ | $0.01 \pm 0.01$ | $0.31 \pm 0.02$ | $0.80 \pm 0.06$ | $0.59 \pm 0.09$ | $0.66 \pm 0.08$ |
| small | avg | centroids | $0.65 \pm 0.13$ | $0.35 \pm 0.05$ | $0.45 \pm 0.05$ | $0.90 \pm 0.05$ | $0.01 \pm 0.01$ | $0.30 \pm 0.02$ | $0.79 \pm 0.06$ | $0.55 \pm 0.10$ | $0.63 \pm 0.08$ |
| small | avg | hard | $0.68 \pm 0.15$ | $0.33 \pm 0.07$ | $0.45 \pm 0.07$ | $0.89 \pm 0.06$ | $0.01 \pm 0.01$ | $0.30 \pm 0.02$ | $0.79 \pm 0.07$ | $0.57 \pm 0.09$ | $0.65 \pm 0.07$ |
| **small** | **mlp** | **basic** | $\mathbf{0.77 \pm 0.11}$ | $\mathbf{0.32 \pm 0.08}$ | $\mathbf{0.47 \pm 0.06}$ | $\mathbf{0.90 \pm 0.05}$ | $\mathbf{0.01 \pm 0.01}$ | $\mathbf{0.31 \pm 0.02}$ | $\mathbf{0.80 \pm 0.07}$ | $\mathbf{0.58 \pm 0.09}$ | $\mathbf{0.65 \pm 0.07}$ |
| small | mlp | centroids | $0.71 \pm 0.13$ | $0.33 \pm 0.06$ | $0.46 \pm 0.05$ | $0.90 \pm 0.05$ | $0.01 \pm 0.01$ | $0.31 \pm 0.02$ | $0.80 \pm 0.06$ | $0.54 \pm 0.09$ | $0.63 \pm 0.08$ |
| small | mlp | hard | $0.84 \pm 0.08$ | $0.25 \pm 0.07$ | $0.44 \pm 0.05$ | $0.90 \pm 0.06$ | $0.01 \pm 0.01$ | $0.31 \pm 0.02$ | $0.79 \pm 0.08$ | $0.58 \pm 0.09$ | $0.65 \pm 0.07$ |

In the previous sections, we showed results using a particular model setup. However, as described in Section 4.3, we also experimented with varying several hyperparameters in our model. Table 3 shows the effect of varying model hyperparameters on the performance of the proposed augmented training scheme as well as the partially- and fully-supervised baseline models. The bold row indicates the model hyperparameters used in Table 1 and Figure 4.

The performance of the baseline models is relatively constant across these hyperparameters. Although the baseline models do not make use of the combination operator directly when during the calibration stage with the unseen test subject, the hyperparameters included here influence how the encoder $F_{\theta_F}$ is trained, and therefore may have an effect on the baselines. The partially-supervised model's constant performance indicates that, despite changes in the pretraining procedure, extrapolating from single gestures to combination gestures remains challenging. The fully-supervised model's constant performance indicates that the feature space continues to be amenable to training a fresh classifier for the unseen subject.

Examining the performance of the model with augmented supervision, there is a clear trade-off between performance on single gestures and performance on combination gestures. The choice of architecture for the auxiliary classifier $G_{\theta_G}^{Pre}$ and the choice of feature combination operator $C_{\theta_C}$ affect this trade-off strongly, while the loss function type has relatively less effect on performance. In all cases, however, the overall accuracy of the augmented model is much greater than the partial supervision model.

### A.2.2 Effect on Feature-Space Similarity

In Table 4, we show how changing model hyperparameters affects the structure of the learned feature space, as measured using the similarity matrix $M_{Sim}$. As described in Section 4.2, we perform pretraining, then extract features of real and synthetic gesture combinations for an unseen test subject, analyze their structure by constructing $M_{Sim}$.

In order to show key information from this large matrix for many scenarios, we summarize $M_{Sim}$ with four key values of interest:

- within-class similarity for real combination gestures ($\texttt{SetSim}(\mathcal{Z}_{comb}, \mathcal{Z}_{comb})$),

- within-class similarity for synthetic combination gestures ($\texttt{SetSim}(\tilde{\mathcal{Z}}_{comb}, \tilde{\mathcal{Z}}_{comb})$),

Table 4: Effect of varying model hyperparameters on feature-space similarity of combination gestures. See Section 4.2 for description of similarity metrics. $G_{\theta_G}^{Pre}$ type; structure of auxiliary classifier used during pretraining. $C_{\theta_C}$ type; structure of combination operator. Triplet type; loss function for pretraining. $\texttt{SetSim}(\mathcal{Z}_{comb}, \mathcal{Z}_{comb})$; average similarity between real items of same class (first 16 diagonal elements of $M_{Sim}$). $\texttt{SetSim}(\tilde{\mathcal{Z}}_{comb}, \tilde{\mathcal{Z}}_{comb})$; average similarity between synthetic items of same class (last 16 diagonal elements of $M_{Sim}$). $\texttt{SetSim}(\mathcal{Z}_{comb}, \tilde{\mathcal{Z}}_{comb})$; average similarity between real and synthetic items of matching label ($16^{th}$ sub-diagonal elements of $M_{Sim}$). $\texttt{SetSim}(\mathcal{Z}_{comb}, \mathcal{Z}'_{comb})$; average similarity between items of non-matching label (all other below-diagonal elements of $M_{Sim}$; note this includes real and synthetic items). Entries show the mean and standard deviation across 50 independent trials. Bold row represents model shown in Table 1 and Figure 4.

| $G_{\theta_G}^{Pre}$ type | $C_{\theta_C}$ type | Triplet type | $\texttt{SetSim}(\mathcal{Z}_{comb}, \mathcal{Z}_{comb})$ | $\texttt{SetSim}(\tilde{\mathcal{Z}}_{comb}, \tilde{\mathcal{Z}}_{comb})$ | $\texttt{SetSim}(\mathcal{Z}_{comb}, \tilde{\mathcal{Z}}_{comb})$ | $\texttt{SetSim}(\mathcal{Z}_{comb}, \mathcal{Z}'_{comb})$ |
|---|---|---|---|---|---|---|
| large | avg | basic | $0.88 \pm 0.03$ | $0.95 \pm 0.01$ | $0.86 \pm 0.02$ | $0.71 \pm 0.02$ |
| large | avg | centroids | $0.12 \pm 0.17$ | $0.20 \pm 0.23$ | $0.06 \pm 0.14$ | $0.02 \pm 0.06$ |
| large | avg | hard | $0.92 \pm 0.06$ | $0.96 \pm 0.03$ | $0.91 \pm 0.06$ | $0.82 \pm 0.13$ |
| large | mlp | basic | $0.71 \pm 0.08$ | $0.82 \pm 0.05$ | $0.62 \pm 0.08$ | $0.38 \pm 0.09$ |
| large | mlp | centroids | $0.10 \pm 0.12$ | $0.17 \pm 0.16$ | $0.03 \pm 0.10$ | $0.01 \pm 0.05$ |
| large | mlp | hard | $0.83 \pm 0.07$ | $0.89 \pm 0.04$ | $0.77 \pm 0.07$ | $0.59 \pm 0.13$ |
| small | avg | basic | $0.36 \pm 0.14$ | $0.56 \pm 0.13$ | $0.27 \pm 0.12$ | $0.10 \pm 0.07$ |
| small | avg | centroids | $0.19 \pm 0.15$ | $0.35 \pm 0.21$ | $0.13 \pm 0.14$ | $0.04 \pm 0.06$ |
| small | avg | hard | $0.64 \pm 0.13$ | $0.79 \pm 0.10$ | $0.58 \pm 0.15$ | $0.34 \pm 0.15$ |
| **small** | **mlp** | **basic** | $\mathbf{0.33 \pm 0.14}$ | $\mathbf{0.51 \pm 0.14}$ | $\mathbf{0.22 \pm 0.12}$ | $\mathbf{0.07 \pm 0.07}$ |
| small | mlp | centroids | $0.17 \pm 0.15$ | $0.31 \pm 0.19$ | $0.10 \pm 0.11$ | $0.03 \pm 0.05$ |
| small | mlp | hard | $0.60 \pm 0.15$ | $0.74 \pm 0.10$ | $0.49 \pm 0.16$ | $0.28 \pm 0.18$ |

- similarity between matching real and synthetic combinations ($\texttt{SetSim}(\mathcal{Z}_{comb}, \tilde{\mathcal{Z}}_{comb})$), and

- similarity between combinations with non-matching labels ($\texttt{SetSim}(\mathcal{Z}_{comb}, \mathcal{Z}'_{comb})$).

The row in bold represents the model hyperparameters used in the main experiments (Section 5).

Note that these feature similarity values are difficult to interpret and compare. The key information comes from comparing the magnitude of similarities for "matching" items against the magnitude of similarities for "non-matching" items; a model whose feature space brings matching items closer together than non-matching items may allow a downstream classifier to more easily learn decision boundaries. A well-structured feature space should have relatively larger values in the first three values (similarity of matching real items, matching synthetic items, and pairs of matching real/synthetic items), and a relatively smaller value in the final column (similarity between non-matching items). However, the absolute magnitude of similarity values on their own is not meaningful, since this only reflects the relative scale of distances in feature space to the kernel lengthscale $\delta$ used in Equation 4. For example, keeping $\delta$ fixed while isotropically shrinking all feature vectors (multiplying all feature vectors by a matrix of the form $\alpha I, \alpha << 1$) would drive all similarity values very close to 1.

Generally, we see that the similarity of non-matching items is lower than for matching items; for most models the non-matching similarity values are many times smaller than the matching similarity values. This is unsurprising, given that we were able to train a downstream classifier successfully for all models. It also gives some circumstantial evidence for the hypothesis that the contrastive learning objective used in pretraining is effective at structuring the feature space, and that this structure correlates with downstream classifier performance.

### A.2.3 Choice of Classifier Algorithm

In Table 5, we show how the balanced accuracy of the unseen subject's final classifier model is affected by the choice of classifier algorithm. In this comparison, all other model hyperparameters were kept constant. We report the mean and standard deviation of balanced accuracy on single gestures, combination gestures, or all gestures, for the model with augmented supervision, the partial supervision baseline, and the full supervision baseline. The first 3 algorithms shown in the table learn a non-linear decision boundary, while

Table 5: Effect of classifier algorithm on balanced accuracy, for the 8 single gesture classes ($Acc_{single}$), the 16 combination gesture classes ($Acc_{comb}$), and all 24 classes ($Acc_{all}$). Augmented supervision; classifier for unseen subject trained according to Section 3.4. Partial Supervision and Full Supervision; baseline models described in Section 4.1. $G_{\theta_G}^{Test}$ Alg; classifier algorithm used for unseen test subject's model. RF; random forest. kNN; k-Nearest Neighbors. LDA; Linear Discriminant Analysis. DT; Decision Tree. LogR; Logistic Regression. All classifiers implemented in Scikit-Learn using default hyperparameters. All models used "small" auxiliary classifier during pretraining, MLP combination operator, and "basic" triplet loss. Entries show the mean and standard deviation across 50 independent trials. Bold row represents model shown in Table 1 and Figure 4.

| $G_{\theta_G}^{Test}$ Alg | Augmented Supervision | | | Partial Supervised | | | Full Supervision | | |
|---|---|---|---|---|---|---|---|---|---|
| | $Acc_{single}$ | $Acc_{comb}$ | $Acc_{all}$ | $Acc_{single}$ | $Acc_{comb}$ | $Acc_{all}$ | $Acc_{single}$ | $Acc_{comb}$ | $Acc_{all}$ |
| **RF** | **0.77 ± 0.11** | **0.32 ± 0.08** | **0.47 ± 0.06** | **0.90 ± 0.05** | **0.01 ± 0.01** | **0.31 ± 0.02** | **0.80 ± 0.07** | **0.58 ± 0.09** | **0.65 ± 0.07** |
| kNN | 0.74 ± 0.12 | 0.33 ± 0.08 | 0.47 ± 0.06 | 0.91 ± 0.05 | 0.00 ± 0.00 | 0.30 ± 0.02 | 0.76 ± 0.07 | 0.57 ± 0.09 | 0.64 ± 0.08 |
| DT | 0.66 ± 0.10 | 0.23 ± 0.06 | 0.38 ± 0.06 | 0.83 ± 0.08 | 0.04 ± 0.03 | 0.31 ± 0.03 | 0.64 ± 0.09 | 0.48 ± 0.07 | 0.53 ± 0.07 |
| LDA | 0.87 ± 0.07 | 0.08 ± 0.05 | 0.35 ± 0.04 | 0.94 ± 0.04 | 0.05 ± 0.03 | 0.35 ± 0.02 | 0.82 ± 0.06 | 0.57 ± 0.09 | 0.66 ± 0.07 |
| LogR | 0.88 ± 0.07 | 0.13 ± 0.06 | 0.38 ± 0.06 | 0.91 ± 0.05 | 0.10 ± 0.04 | 0.37 ± 0.03 | 0.77 ± 0.07 | 0.61 ± 0.08 | 0.66 ± 0.07 |

the the final 2 algorithms can only learn linear decision boundaries. Two main patterns stand out from these results. First, most classification algorithms show the same trend, where the augmented training results in a small decrease in accuracy on single gesture classification, and a large increase in accuracy on combination gesture classification, resulting in a net benefit overall as compared to the partial supervision model. Second, it appears that classification algorithms with a non-linear decision boundary benefit much more from the augmented training, and achieve higher overall accuracy as a result.

## A.3 Ablation Experiments

We conduct ablation experiments on several elements of our pretraining procedure. Specifically, we train models in which one more of the three loss terms in Equation 3 are removed. We also vary the amount of noise added to the raw input data, as described in Equation 6. In these experiments, we use the same model hyperparameters as in Table 1 and Figure 4; specifically, the auxiliary classifier $G_{\theta_G}^{Pre}$ used during pretraining is the "small" architecture, the combination operator $C_{\theta_C}$ is the MLP architecture, and the triplet loss function used is the "basic" version.

### A.3.1 Effect on balanced accuracy

Table 6: Effect of ablations on balanced accuracy, for the 8 single gesture classes ($Acc_{single}$), the 16 combination gesture classes ($Acc_{comb}$), and all 24 classes ($Acc_{all}$). Augmented supervision; classifier for unseen subject trained according to Section 3.4. Partial Supervised and Full Supervision; baseline models described in Section 4.1. $\mathcal{L}_{TRIPLET}$, $\mathcal{L}_{CE}$, $\tilde{\mathcal{L}}_{CE}$; loss terms in Equation 3. $SNR$; signal-to-noise ratio (dB) after adding noise to raw input data. All models used random forest algorithm for unseen test subject, "small" architecture for auxiliary classifier $G_{\theta_G}^{Pre}$, MLP combination operator $C_{\theta_C}$, and "basic" triplet loss. Entries show the mean and standard deviation across 50 independent trials. Bold row represents non-ablated model used in Table 1 and Figure 4.

| Hyperparameters | | | | Augmented Supervision | | | Partial Supervision | | | Full Supervision | | |
|---|---|---|---|---|---|---|---|---|---|---|---|---|
| $\mathcal{L}_{triplet}$ | $\mathcal{L}_{ce}$ | $\tilde{\mathcal{L}}_{ce}$ | $SNR$ | $Acc_{single}$ | $Acc_{comb}$ | $Acc_{all}$ | $Acc_{single}$ | $Acc_{comb}$ | $Acc_{all}$ | $Acc_{single}$ | $Acc_{comb}$ | $Acc_{all}$ |
| - | - | ✓ | 20.0 | 0.71 ± 0.11 | 0.19 ± 0.04 | 0.36 ± 0.04 | 0.86 ± 0.06 | 0.00 ± 0.00 | 0.29 ± 0.02 | 0.71 ± 0.08 | 0.45 ± 0.08 | 0.54 ± 0.08 |
| - | ✓ | - | 20.0 | 0.69 ± 0.14 | 0.32 ± 0.06 | 0.44 ± 0.05 | 0.89 ± 0.06 | 0.01 ± 0.01 | 0.30 ± 0.02 | 0.78 ± 0.08 | 0.58 ± 0.09 | 0.65 ± 0.08 |
| - | ✓ | ✓ | 20.0 | 0.69 ± 0.13 | 0.33 ± 0.06 | 0.45 ± 0.05 | 0.88 ± 0.06 | 0.01 ± 0.01 | 0.30 ± 0.02 | 0.78 ± 0.07 | 0.54 ± 0.10 | 0.62 ± 0.08 |
| ✓ | - | - | 20.0 | 0.85 ± 0.06 | 0.16 ± 0.06 | 0.39 ± 0.05 | 0.87 ± 0.06 | 0.00 ± 0.01 | 0.29 ± 0.02 | 0.76 ± 0.07 | 0.56 ± 0.09 | 0.63 ± 0.08 |
| ✓ | - | ✓ | 20.0 | 0.86 ± 0.06 | 0.23 ± 0.07 | 0.44 ± 0.05 | 0.91 ± 0.04 | 0.01 ± 0.02 | 0.31 ± 0.02 | 0.80 ± 0.06 | 0.62 ± 0.09 | 0.68 ± 0.07 |
| ✓ | ✓ | - | 20.0 | 0.76 ± 0.11 | 0.30 ± 0.07 | 0.45 ± 0.05 | 0.90 ± 0.05 | 0.01 ± 0.01 | 0.30 ± 0.02 | 0.80 ± 0.07 | 0.59 ± 0.10 | 0.66 ± 0.08 |
| ✓ | ✓ | ✓ | 10.0 | 0.81 ± 0.08 | 0.30 ± 0.06 | 0.47 ± 0.05 | 0.91 ± 0.06 | 0.01 ± 0.01 | 0.31 ± 0.02 | 0.81 ± 0.07 | 0.61 ± 0.08 | 0.67 ± 0.07 |
| ✓ | ✓ | ✓ | 30.0 | 0.75 ± 0.11 | 0.33 ± 0.07 | 0.47 ± 0.05 | 0.90 ± 0.05 | 0.01 ± 0.01 | 0.31 ± 0.02 | 0.79 ± 0.06 | 0.56 ± 0.08 | 0.64 ± 0.07 |
| ✓ | ✓ | ✓ | ∞ | 0.74 ± 0.12 | 0.34 ± 0.06 | 0.47 ± 0.05 | 0.90 ± 0.05 | 0.01 ± 0.01 | 0.31 ± 0.02 | 0.80 ± 0.07 | 0.57 ± 0.09 | 0.65 ± 0.07 |
| **✓** | **✓** | **✓** | **20.0** | **0.77 ± 0.11** | **0.32 ± 0.08** | **0.47 ± 0.06** | **0.90 ± 0.05** | **0.01 ± 0.01** | **0.31 ± 0.02** | **0.80 ± 0.07** | **0.58 ± 0.09** | **0.65 ± 0.07** |

In Table 6, we show the balanced accuracy of various ablated models. We find that the original, non-ablated model (shown in bold) performs best. Of the three loss terms, the best single term to include is the real cross-entropy $\mathcal{L}_{\text{CE}}$; surprisingly, including only the triplet loss $\mathcal{L}_{\text{TRIPLET}}$ results in a model that performs better on single gestures than on combination gestures.

### A.3.2 Effect on Feature-Space Similarity

Table 7: Effect of model ablations on feature-space similarity of combination gestures. See Section 4.2 for description of similarity metrics. $\mathcal{L}_{\text{TRIPLET}}$, $\mathcal{L}_{\text{CE}}$, $\tilde{\mathcal{L}}_{\text{CE}}$; loss terms in Equation 3. $SNR$; signal-to-noise ratio (dB) after adding noise to raw input data. $\texttt{SetSim}(\mathcal{Z}_{comb}, \mathcal{Z}_{comb})$; average similarity between real items of same class (first 16 diagonal elements of $M_{Sim}$). $\texttt{SetSim}(\tilde{\mathcal{Z}}_{comb}, \tilde{\mathcal{Z}}_{comb})$; average similarity between synthetic items of same class (last 16 diagonal elements of $M_{Sim}$). $\texttt{SetSim}(\mathcal{Z}_{comb}, \tilde{\mathcal{Z}}_{comb})$; average similarity between real and synthetic items of matching label ($16^{th}$ sub-diagonal elements of $M_{Sim}$). $\texttt{SetSim}(\mathcal{Z}_{comb}, \mathcal{Z}'_{comb})$; average similarity between items of non-matching label (all other below-diagonal elements of $M_{Sim}$; note this includes real and synthetic items). Entries show the mean and standard deviation across 50 independent trials. Bold row represents model shown in Table 1 and Figure 4.

| Hyperparameters | | | | | | | |
|:---:|:---:|:---:|:---:|:---:|:---:|:---:|:---:|
| $\mathcal{L}_{\text{triplet}}$ | $\mathcal{L}_{\text{ce}}$ | $\tilde{\mathcal{L}}_{\text{ce}}$ | $SNR$ | $\texttt{SetSim}(\mathcal{Z}_{comb}, \mathcal{Z}_{comb})$ | $\texttt{SetSim}(\tilde{\mathcal{Z}}_{comb}, \tilde{\mathcal{Z}}_{comb})$ | $\texttt{SetSim}(\mathcal{Z}_{comb}, \tilde{\mathcal{Z}}_{comb})$ | $\texttt{SetSim}(\mathcal{Z}_{comb}, \mathcal{Z}'_{comb})$ |
| - | - | ✓ | 20.0 | $0.28 \pm 0.11$ | $0.41 \pm 0.13$ | $0.15 \pm 0.08$ | $0.05 \pm 0.04$ |
| - | ✓ | - | 20.0 | $0.21 \pm 0.18$ | $0.39 \pm 0.20$ | $0.12 \pm 0.12$ | $0.06 \pm 0.07$ |
| - | ✓ | ✓ | 20.0 | $0.19 \pm 0.19$ | $0.36 \pm 0.21$ | $0.13 \pm 0.15$ | $0.06 \pm 0.09$ |
| ✓ | - | - | 20.0 | $0.92 \pm 0.03$ | $0.94 \pm 0.01$ | $0.85 \pm 0.03$ | $0.72 \pm 0.07$ |
| ✓ | - | ✓ | 20.0 | $0.65 \pm 0.09$ | $0.78 \pm 0.05$ | $0.53 \pm 0.08$ | $0.26 \pm 0.08$ |
| ✓ | ✓ | - | 20.0 | $0.36 \pm 0.14$ | $0.52 \pm 0.13$ | $0.23 \pm 0.11$ | $0.09 \pm 0.06$ |
| ✓ | ✓ | ✓ | 10.0 | $0.42 \pm 0.09$ | $0.60 \pm 0.08$ | $0.28 \pm 0.08$ | $0.11 \pm 0.05$ |
| ✓ | ✓ | ✓ | 30.0 | $0.31 \pm 0.18$ | $0.48 \pm 0.17$ | $0.21 \pm 0.16$ | $0.08 \pm 0.09$ |
| ✓ | ✓ | ✓ | $\infty$ | $0.30 \pm 0.16$ | $0.47 \pm 0.16$ | $0.20 \pm 0.13$ | $0.07 \pm 0.06$ |
| **✓** | **✓** | **✓** | **20** | $\mathbf{0.33 \pm 0.14}$ | $\mathbf{0.51 \pm 0.14}$ | $\mathbf{0.22 \pm 0.12}$ | $\mathbf{0.07 \pm 0.07}$ |

As mentioned previously, we used a fixed kernel lengthscale $\delta$ for all experiments. When interpreting similarity values, the key outcome we desire is that matching classes should have higher similarity values than non-matching classes. In general, we observe that this remains true across all model ablations, though the degree of contrast varies. Note that due to the dynamics of gradient descent during the pretraining stage, it is possible that some model's feature spaces have a generally shorter or longer lengthscale, and this may affect the apparent contrast in similarity values.

### A.3.3 Relative Loss Scaling

In Equation 3, we define a multi-objective optimization. The magnitudes of the triplet loss and cross-entropy losses may not be on the same scale, since the triplet loss depends on a margin parameter, and on the scaling of the feature space. Thus, we consider a modified objective with an additional coefficient to scale the relative size of these two types of losses:

$$\mathcal{L} = \mathcal{L}_{\text{TRIPLET}} + \alpha(\mathcal{L}_{\text{CE}} + \tilde{\mathcal{L}}_{\text{CE}}). \tag{7}$$

We vary the coefficient $\alpha$ and measure the resulting balanced accuracy of the model as before. We find that varying the relative magnitude of triplet and cross-entropy losses slightly changes the trade-off between single and combination gesture performance. Specifically, we observe that increasing $\alpha$ in Eq. (7) causes a decrease in $Acc_{single}$ and an increase in $Acc_{comb}$. However, this change in the trade-off does not provide an increase in $Acc_{all}$, indicating other techniques may be required to achieve better performance on both single and combination classes simultaneously.

Table 8: Effect of varying relative loss scaling on balanced accuracy for the 8 single gesture classes ($Acc_{single}$), the 16 combination gesture classes ($Acc_{comb}$), and all 24 classes ($Acc_{all}$). Augmented supervision; classifier for unseen subject trained according to Section 3.4. Partial Supervised and Full Supervision; baseline models described in Section 4.1. A single coefficient $\alpha$ scales the relative magnitude of triplet and cross-entropy losses in Eq. (7). Changing $\alpha$ changes the trade-off in performance on single and combination gestures, but does not affect overall accuracy across all classes. Entries show the mean and standard deviation across 50 independent trials. Bold row represents the model shown in Table 1 and Figure 4.

| Hyperparameter $\alpha$ in Eq. (7) | Augmented Supervision | | | Partial Supervision | | | Full Supervision | | |
|---|---|---|---|---|---|---|---|---|---|
| | $Acc_{single}$ | $Acc_{comb}$ | $Acc_{all}$ | $Acc_{single}$ | $Acc_{comb}$ | $Acc_{all}$ | $Acc_{single}$ | $Acc_{comb}$ | $Acc_{all}$ |
| 0.1 | $0.86 \pm 0.08$ | $0.25 \pm 0.07$ | $0.45 \pm 0.05$ | $0.91 \pm 0.05$ | $0.01 \pm 0.01$ | $0.31 \pm 0.02$ | $0.81 \pm 0.07$ | $0.64 \pm 0.09$ | $0.69 \pm 0.08$ |
| 0.5 | $0.80 \pm 0.10$ | $0.31 \pm 0.06$ | $0.47 \pm 0.05$ | $0.90 \pm 0.06$ | $0.01 \pm 0.01$ | $0.30 \pm 0.02$ | $0.80 \pm 0.07$ | $0.61 \pm 0.08$ | $0.67 \pm 0.06$ |
| **1.0** | $\mathbf{0.77 \pm 0.11}$ | $\mathbf{0.32 \pm 0.08}$ | $\mathbf{0.47 \pm 0.06}$ | $\mathbf{0.90 \pm 0.05}$ | $\mathbf{0.01 \pm 0.01}$ | $\mathbf{0.31 \pm 0.02}$ | $\mathbf{0.80 \pm 0.07}$ | $\mathbf{0.58 \pm 0.09}$ | $\mathbf{0.65 \pm 0.07}$ |
| 2.0 | $0.73 \pm 0.12$ | $0.34 \pm 0.07$ | $0.47 \pm 0.06$ | $0.89 \pm 0.05$ | $0.01 \pm 0.01$ | $0.30 \pm 0.02$ | $0.79 \pm 0.07$ | $0.57 \pm 0.10$ | $0.64 \pm 0.08$ |
| 10.0 | $0.71 \pm 0.13$ | $0.36 \pm 0.06$ | $0.47 \pm 0.05$ | $0.89 \pm 0.05$ | $0.01 \pm 0.01$ | $0.30 \pm 0.02$ | $0.79 \pm 0.07$ | $0.57 \pm 0.10$ | $0.64 \pm 0.09$ |

