# OpenReview forum: "Fast and Expressive Gesture Recognition using a Combination-Homomorphic Electromyogram Encoder"
_TMLR — Accepted by TMLR_

### Review · Reviewer_Um9S · 2024-01-15

**Summary Of Contributions:**

This paper is about training a classifier with individual gesture samples (gesture + modifier), and basically being able to generalize to combinations of those gestures (cross product of gesture and modifier, gesture x modifier) , by performing contrastive learning to learn features that are similar in both setups.

The paper designs and evaluates this method specifically for EMG gesture recognition, where the authors captured EMG data with users performing certain gestures, both in single and combined settings, and the proposed method is trained on the single setting, while evaluated on the combined setting, allowing a degree of generalization outside of the training set.

Results show that the proposed method using a contrastive learning pipeline, outperforms the baseline (which does not generalize in the combined setting) while at the same time having a reduction in performance in the single setting.

The contributions of this paper are:
- A method for generalization for gesture recognition, where classes can be divided into single class, or combinations of classes (similar to a multi-label setting).
- Using partial supervision for the problem of gesture recognition in single/combined settings.
- Results that indicate that the proposed method works better than naive solutions, better generalizing in the combined setting, while reducing performance in the single setting.

**Audience:**

Yes

**Broader Impact Concerns:**

Broader impact looks good t o me, no additional ethical implications.

**Claims And Evidence:**

No

**Requested Changes:**

- It would be very nice if the claims in the paper are explicitly written at the end of the introduction, same for the contributions, in order to make sure authors and reviewers are on the same page about what is being claimed.
- I would expect that the authors make a deeper dive into why the single setting performance reduces when using the triplet loss and partial supervision, while I do not expect or require new experiments, this can probably be answered with some experimentation, but I am not sure.
- Also please consider adding loss weights for Eq 3 and ablation experiments for those loss weights.

**Strengths And Weaknesses:**

Strengths
- The paper is nicely written, most details are clear. Figures are high quality and convey the complexity of the problem and the solution.
- The problem description is clear, it is a complicated problem, due to having multiple settings (single and combined), not common in a machine learning problem, where classes are known a priori and there is no expectation of generalization to combinations of classes not seen during training.
- The problem is very interesting for the machine learning community, generalizing for partial data is an extremely interesting problem as it can impact/lessen the amount of data required for generalization, and overall generalizing from single classes to combination of classes is an extremely interesting problem from a supervision perspective. From this perspective the treatment of this problem is a contribution to the state of the art.
- I believe the evaluation is correct, there is a good selection of metrics, only a single dataset due to the uniqueness of the problem, and three different supervision settings, with progressively more supervision added to the problem.
- The overall result is that the proposed augmented supervision method improves on partial supervision (training on single settings while evaluating on the combined setting), with balanced accuracy improving from 1% to 32% (over 16+1 classes), while reducing accuracy in the single setting from 90% to 77%.
- Overall I believe this paper proposes a novel method for a novel problem, and is a good contribution to the state of the art that is significant for the field and audience of machine learning.

Weaknesses
- Something a bit concerning is that the proposed method produces a reduction in balanced accuracy in the single setting, together with an improvement in the combined setting. This is fine, but the paper does not dive deeper on why there is a reduction in the single setting, at least as validation this should be done, as it might point to future research or at least could be an actionable weakness.
- I believe some ablation studies could have been performed, for example, in Equation 3, weights should been added to this multi-task loss as the triplet and cross-entropy losses are not in the same scale, and an ablation study could be made for the loss weights. Maybe this way some trade-off between single and combined case performance can be found (this is just speculation).

---

> ### Author Response · Authors · 2024-03-08
> **Response to Review**
>
> > It would be very nice if the claims in the paper are explicitly written at the end of the introduction, same for the contributions, in order to make sure authors and reviewers are on the same page about what is being claimed.
>
> We have added an explicit list of contributions in the Introduction, and also expanded the Discussion section to give more context on the claims being made and to ensure our claims are well-supported.
>
> > Something a bit concerning is that the proposed method produces a reduction in balanced accuracy in the single setting, together with an improvement in the combined setting. This is fine, but the paper does not dive deeper on why there is a reduction in the single setting, at least as validation this should be done, as it might point to future research or at least could be an actionable weakness. ... I would expect that the authors make a deeper dive into why the single setting performance reduces when using the triplet loss and partial supervision, while I do not expect or require new experiments, this can probably be answered with some experimentation, but I am not sure.
>
> We agree that this accuracy trade-off between single and combination labels is an interesting phenomenon and warrants further discussion. We have added a new section and experiment (new Table 2) to examine and explain this phenomenon, which we briefly summarize here.
>
> The key phenomenon is that the full data distribution is multi-modal. The partially-supervised model is trained to fit only the mode coming from single gestures. As a result, the partially-supervised model performs very well on single gestures but poorly elsewhere.
>
> In the new experiment, we use the ability to classify different subsets of data as an indirect measure of separation between these subsets. This lets us demonstrate that our raw data is multi-modal as we claim. We train a Random Forest classifier on raw data for the following tasks:
> - Predict whether a gesture is single ("D, NoMod") or ("NoDir, M") vs combination ("D, M")
> - Predict which direction is active ("Up, Any") vs ("Down, Any") vs ("Left, Any") vs ("Right, Any")
> - Predict which modifier is active ("Any, Pinch") vs ("Any, Thumb") vs ("Any, Open") vs ("Any, Fist")
>
> We find that it is roughly as easy to distinguish single and combination data (73.2%), as it is to distinguish between the four active direction gestures (74.7%), or to distinguish between the four active modifier gestures (66.7%).
>
> > I believe some ablation studies could have been performed, for example, in Equation 3, weights should been added to this multi-task loss as the triplet and cross-entropy losses are not in the same scale, and an ablation study could be made for the loss weights. Maybe this way some trade-off between single and combined case performance can be found (this is just speculation). ... Also please consider adding loss weights for Eq 3 and ablation experiments for those loss weights.
>
> We have added an additional experiment at the end of the Appendix section in which we vary the relative magnitude of triplet and cross-entropy losses. Note that we already included a binary ablation (where each loss was multiplied by $0$ or $1$ in various combinations), and several other ablations in the Appendix.
> The results of this new experiment are consistent with the previous binary ablation. We find that varying the relative magnitude changes the trade-off between $Acc_{single}$ and $Acc_{comb}$, but does not benefit $Acc_{all}$. Specifically, when cross-entropy terms are down-weighted, the model has higher $Acc_{single}$, and when cross-entropy terms are up-weighted, the model has higher $Acc_{comb}$.

---

### Review · Reviewer_oc36 · 2024-01-16

**Summary Of Contributions:**

The paper focuses on the problem of efficient training of EEG-based gesture recognition models on unseen subjects. The gestures are defines as combinations of “directions” and “modifiers”, which makes collecting all pairs of data time-consuming and labor-intensive. Therefore, this paper aims to only collect either direction or modifier labels to train the gesture recogntion model. To achieve this, it proposes a self-supervised contrastive learning approach, where synthetic combinations of gesture labels are created, and a triplet loss is used to learn a feature encoder that maps EEG of real gesture pairs to features that are far away from fake gesture pairs. Once the encoder is learned, it is used to train a gesture classifier, which is expected to have higher accuracy since it can combine the “direction” and “modifier” features better. Experiments show that the proposed augmented supervision approach outperforms baselines that only use partial supervision w/ either gesture direction or modifier for training.

**Audience:**

Yes

**Broader Impact Concerns:**

- The paper follows good IRB protocols and the research like doesn't present specific broader impact concerns.

**Claims And Evidence:**

Yes

**Requested Changes:**

- More/stronger baselines to show the method advances the state of the art.
- More tasks to demonstrate the generality of the proposed method.

**Strengths And Weaknesses:**

**Strengths:**

- The paper does a good job of introducing the problem and motivating why learning a combination-homomorphic encoder is important and useful.
- The method design makes sense, i.e., contrastive learning to encourage encoded features to be close to real gesture pairs and far away from fake pairs. This also looks novel to me, although I am not familiar with self-supervised/contrastive learning literature.
- Experiments show that the proposed method can largely increase the model performance on real pairs of gestures compared to the baselines.
- The paper is well-written and easy to follow.

**Weaknesses:**

- The partial supervision baseline seems too simple and easy to beat. I’m sure there are self-supervised methods to learn better representations using real paired data. E.g., training a VAE w/ an auxiliary classification loss.
- Currently, the method is only validated on a single task and single datasets. Since the method is general for combination labels, I’d like to see it being applied to more tasks (even if it’s toy/synthetic examples).

---

> ### Author Response · Authors · 2024-03-08
> **Response to Review**
>
> > The partial supervision baseline seems too simple and easy to beat. I’m sure there are self-supervised methods to learn better representations using real paired data. E.g., training a VAE w/ an auxiliary classification loss. ... More/stronger baselines to show the method advances the state of the art.
>
> We consider our chosen baseline models appropriate for answering our main experimental question, which is: _Can synthetic combinations substitute for real combinations in a domain adaptation problem with a combinatorial label structure?_
>
> We have expanded the discussion section to give more context on our experimental design and choice of baselines, and we summarize here.
>
> Recall that we focus on using a structured form of partial supervision to reduce calibration time, and compensate for this reduced supervision using contrastive learning. Our experiments show that data synthesis (or real combination data) is necessary during model calibration.
>
> While self-supervised methods such as VAEs are able to learn informative features, we believe that they would not learn an arrangement of classes that allows data synthesis, since the effect of the prior loss in a VAE is to force the feature distribution (across all classes) closer to a Gaussian. Adding auxiliary classification losses ensures that features can be classified, but still does not ensure they can be combined into realistic synthetic examples. For example, each class could still be mapped to multiple disjoint regions of feature space, and need not be placed well for making combinations.
>
> We also emphasize a few points about our experimental design. The partial supervision baseline we considered is designed to evaluate whether the synthetic examples that we produce are useful for model calibration. It shows how classification would proceed when using no labeled combination gestures for calibration. The full supervision model represents performance after an exhaustive calibration session with all classes, and captures the effect of factors such as data noise, model architecture, and classifier algorithm.
>
> > Currently, the method is only validated on a single task and single datasets. Since the method is general for combination labels, I’d like to see it being applied to more tasks (even if it’s toy/synthetic examples). ... More tasks to demonstrate the generality of the proposed method.
>
> We focused particularly on extrapolating from single labels to unseen combinations. We agree that there is an interesting potential to apply the proposed method more generally, and expanded the discussion section to include this topic.
>
> The scope of this paper is strongly motivated by the specific task that we consider, which is calibrating EMG classification systems to new users from limited calibration data.
> The general task setting may be considered as a few-shot learning or transfer-learning task, in a domain with a combinatorial label structure.
> Although many real-world problems match this setting, there are limited existing benchmarks with these features.
> We consider the dataset collected to be one of our main contributions (the dataset will be made publicly available upon publication of our manuscript).
>
> We have expanded the discussion section to describe potential future applications and related tasks.
>
> One area is tasks with multiple rare labels; since there may be insufficient examples of combination data, synthesizing combinations could be useful. Note that our proposed approach relied on a fully-supervised pre-training, and thus may require adjustments if real combination data are unavailable during pre-training.
>
> Another area is tasks with many labeled attributes, where the total space of combinations is too large to explore, such as in scene understanding and autonomous navigation. Here, synthesizing examples may be useful since collecting real examples from all combination classes may be infeasible.
>
> We agree that well-designed toy datasets would provide an interesting test case by
> allowing us to explore the effect of different combination functions and the ability to handle different class structures. We note that producing a useful toy dataset for this problem would require significant work and may be out-of-scope for the current study.

---

### Review · Reviewer_aZhy · 2024-02-23

**Summary Of Contributions:**

This paper targets the task of gesture recognition from electromyography (EMG) as a machine learning application and to improve the accuracy while minimizing the calibration cost for adapting the model to a new subject. The authors consider the combination gestures consisting of direction and modifier components and the situation where only the single component gestures are given for a new subject. To extrapolate unseen combination gestures, the authors generate synthetic training data from real single gestures. The authors propose a novel encoder pre-trining strategy to produce useful synthetic training data for unseen test subjects and to possess the homomorphisms of combining feature vectors. In the experiment, the effectiveness of the proposed method is evaluated using a real EMG dataset. The experimental result demonstrates that the proposed method can improve the recognition accuracy compared to the partially-supervised model.

**Audience:**

Yes

**Broader Impact Concerns:**

I do not have an additional concern on the ethical implications of this paper.

**Claims And Evidence:**

Yes

**Requested Changes:**

1. Does the proposed method work well on other datasets or tasks other than that used in this paper? Could you discuss the generality of the proposed method and the possible applications other than gesture recognition from EMG?
1. I am not sure whether the accuracies (e.g., $Acc_{single}=0.77$ and $Acc_{comb}=0.32$) of the proposed method are sufficient for a target application. It would be better to explain the accuracy requirements from the viewpoint of the target application.
1. I am not sure whether the considered baseline methods are appropriate. Can we consider the domain adaptation and few-shot learning methods as baseline methods? Could you elaborate on the justification for the selection of the baseline methods?

[Minor comments]
- In Section 1: "Noise in EMG data can stem from from the..." -> "Noise in EMG data can stem from the..."

**Strengths And Weaknesses:**

[Strengths]
- The task and problem definition of subject transfer learning is well-motivated and described.
- The proposed pre-training method of an EMG encoder is well-designed. The use of triplet loss is convincing.

[Weaknesses]
- The experimental evaluation is conducted on only one dataset collected by the authors.
- The baseline methods compared to the proposed method might be too simple.

---

> ### Author Response · Authors · 2024-03-08
> **Response to Review**
>
> > Does the proposed method work well on other datasets or tasks other than that used in this paper? Could you discuss the generality of the proposed method and the possible applications other than gesture recognition from EMG?
>
> The proposed method may work well on other datasets, though we do not directly test this. Our method was developed specifically to address the challenging extrapolation properties of this EMG classification task.  In addition to developing a method suitable for transferring to new users with limited calibration data, another significant contribution of this work was collecting and releasing a real-world dataset of labeled EMG data.
>
> We agree our method may be extended for other future tasks.  One area is tasks with multiple rare labels, such as classifying electrocardiogram data in which multiple rare disease signature can co-occur. There, a model must classify these rare events well to be useful, but there may not be enough real examples of co-occurrence, so synthetic supervision may be useful. Our method would likely need adaptation, since we use a fully-supervised pretraining in order to learn to create useful synthetic data.
>
> Another area is tasks with many attributes where the space of labels is too large to fully explore, such as robotic learning and world modeling. Here, synthetic supervision could be used for unseen combinations, though our proposed method may need adjustment: we collect a full "basis set" of single gestures that can describe all possible combinations, but that setting may require combining more than two examples, or a multi-step combination procedure such as adding some feature vectors and subtracting others.
>
> We have added more context and citations to explain other potential applications and the relationship with tasks in domain adaptation and compositional generalization.
> > I am not sure whether the accuracies (e.g., $Acc_{single} = 0.77$ and $Acc_{comb} = 0.32$) of the proposed method are sufficient for a target application. It would be better to explain the accuracy requirements from the viewpoint of the target application.
>
> We have added a section to the discussion examining our current results from the perspective of a user-facing application, and summarize here.
>
> A per-window accuracy of 77% on single gestures suggests our model may be part of a useful real-world system. However, 32% is probably insufficient without additional techniques, or introducing a small amount of real labeled calibration data.
>
> In a user-facing application, an underlying model such as ours is typically used alongside several aggregation steps to achieve higher accuracy:
> - Aggregating predictions from multiple consecutive data windows, e.g. https://arxiv.org/pdf/2203.15239.pdf for gesture recognition from IMU data
> - Using a task-specific Bayesian prior, e.g. https://www.ncbi.nlm.nih.gov/pmc/articles/PMC5495144/ for assistive typing via electroencephalography
> - Using an ensemble of model copies, e.g. https://proceedings.mlr.press/v162/du22c/du22c.pdf for language modeling.
>
> Implementing these steps is conceptually simple, but requires a non-trivial amount of work, and also relies on defining and implementing an application scenario (such as a typing task, or a user interface task) to collect evaluation metrics. We defer this type of extension for future research.
>
> Note that even the fully-supervised baseline model only achieves 58% on combination gestures, indicating performance is limited by other factors such as noise in the data, limited amount of training data, and choices of classifier algorithm.
> > I am not sure whether the considered baseline methods are appropriate. Can we consider the domain adaptation and few-shot learning methods as baseline methods? Could you elaborate on the justification for the selection of the baseline methods?
>
> We consider the choice of baselines suitable to answer our main question, which is: _Can synthetic combinations substitute for real combinations in a domain adaptation problem with a combinatorial label structure?_ We have expanded the Discussion to give more context for our experimental design.
>
> Our task is motivated by reducing time for end users to calibrate a personalized model, and contains aspects of both domain adaptation and combinatorial generalization. Domain adaptation or few-shot learning methods that cannot address both challenges are unlikely to  perform well on this task alone. However, our method may be used along with other adaptation methods.
>
> Our choice of baselines is designed to measure the effect of synthetic combinations for model calibration. The partial-supervision model trains using the same real data as our method, and demonstrates performance using no labeled combinations for calibration. The full supervision model is trained with additional data of real combinations. The goal of comparing to this model is to show how close training with synthetic combinations can get to training with real data.

---

### Author Response · Authors · 2024-03-08
**Summary of Changes**

We thank the reviewers for their time, attention, and detailed feedback. It is clear that the reviewers understood our work well, and we feel that the changes made in response have greatly improved the quality of our manuscript.

Here we summarize some important changes:
- We expanded the Discussion section to give context on many aspects of our study, including: our experimental design and choice of baselines (R1 - aZhy, R2 - oc36), the trade-off between performance on single gestures and combination gestures (R3 - Um9S), the relationship between our task and related tasks (R1 - aZhy, R2 - oc36), examining our performance in the context of a user-facing application (R1 - aZhy).
- We added a "contributions" section that makes the claims of our paper more clear and explicit. We only claim points that are well-supported by our experiments. (R3 - Um9S)
- We added a classification experiment on the raw data to show that the single and combination data are substantially separated. (R3 - Um9S)
- We added an ablation experiment to vary the magnitude of terms in our loss function. (R3 - Um9S)

---

### Decision · Action_Editor_BKjE · 2024-03-31

**Recommendation:** Accept as is

**Comment:**

The paper focuses on the problem of efficient training of EEG-based gesture recognition models on unseen subjects. The task of subject transfer learning is well-motivated and has good applications. After the first round of reviews, there are some major concerns on analysis of the single case and ablation studies, the generality of the proposed method and experimental comparisons.  These issues have been well addressed in the revision and all reviewers are satisfied with the revision.

**Audience:**

The task of subject transfer learning is well-motivated and has good applications, which would be interested in the community.

**Claims And Evidence:**

In the first round of reviews, although several concerns regarding the generality of the proposed method and experimental comparisons were raised, these concerns are addressed in the revised paper.  The main claims are well supported by convincing and clear evidence.